# Unlocking Graph Structure Learning with Tree-Guided Large Language Models

## Abstract

Recently, the emergence of large language models (LLMs) has prompted researchers to integrate language descriptions into graphs, aiming to enhance model encoding capabilities from a data-centric perspective. This graph representation is called text-attributed graphs (TAGs). A review of prior advancements highlights that graph structure learning (GSL) is a pivotal technique for improving data utility, making it highly relevant to efficient TAG learning. However, most GSL methods are tailored for traditional graphs without textual information, underscoring the necessity of developing a new GSL paradigm. Despite clear motivations, it remains challenging: (1) How can we define a reasonable optimization objective for GSL in the era of LLMs, considering the massive parameters in LLMs? (2) How can we design an efficient model architecture that enables seamless integration of LLMs for this optimization objective? For Question 1, we reformulate existing GSL optimization objectives as a tree optimization framework, shifting the focus from obtaining a well-trained edge predictor to a language-aware tree sampler. For Question 2, we propose decoupled and training-free model design principles for LLM integration, shifting the focus from computation-intensive fine-tuning to more efficient inference. Based on this, we propose Large Language and Tree Assistant (LLaTA), which leverages tree-based LLM in-context learning to enhance the understanding of topology and text, enabling reliable inference and generating improved graph structure. Extensive experiments on 11 datasets demonstrate that LLaTA enjoys flexibility—incorporated with any backbone; scalability—outperforms other LLM-based GSL methods; and effectiveness—achieving SOTA predictive performance across a variety of datasets from different domains.

## 1 Introduction

In recent years, the rise of LLMs in graph ML Yan et al. (2023); Chen et al. (2024) has led to the emergence of text-attributed graphs (TAGs) as a novel data representation. This data structure leverages textual information to provide fine-grained descriptions of graphs, driving a significant shift in data-centric graph learning Li et al. (2024). Inspired by this, efficiently and robustly learning TAGs has become an urgent priority.

To achieve this, graph structure learning (GSL) is a promising approach to enhancing data utility for vanilla graph neural networks (GNNs). However, most existing GSL methods Zhiyao et al. (2024) are designed for traditional graphs and cannot effectively process rich textual information, leading to suboptimal performance. Despite recent efforts in LLM-based GSL approaches, such as GraphEdit Guo et al. (2024) and LLM4RGNN Zhang et al. (2024), their optimization objectives and model architectures still follow traditional GSL paradigms, inheriting unnecessary complexities. Specifically, these prominent GSL methods Zhao et al. (2021); Wu et al. (2022b; 2023); Zhang et al. (2024) rely on carefully designed loss functions to maintain a graph learner (i.e., structure optimizer) coupled with well-defined instruction datasets or specific GNN backbones (i.e., improved structure tailored for the downstream encoder). Based on this, we make two key observations and aim to explore them to guide the development of a new GSL paradigm in the era of LLMs.

**Observation 1** (Optimization Objective): The graph learner often serves as an edge predictor, heavily relying on end-to-end training with a specific downstream backbone. In the era of LLMs, LLMs will play a crucial role in graph learners. In this context, performing full-parameter LLM training to obtain an edge predictor is infeasible. While two recent fine-tuning methods have been proposed, the complexity of constructing instruction datasets and tuning still hinders their efficiency. This leads to **Question 1**: How can we define a reasonable optimization objective for LLM-enhanced GSL?

**Answer 1**: In Sec. 3.1, we present a new *optimization framework* that reformulates the existing GSL optimization objective (well-trained edge predictor) as a tree-based optimization task (well-defined language-aware tree sampler) to achieve topology-preserving structural optimization.

**Observation 2** (Model Architecture): To obtain a well-trained edge predictor, the existing graph learner is often coupled with customized instruction datasets or a specific graph learning backbone, with gradient supervision for model parameter updates in a collaborative manner. The complexity of this model architecture hinders the adaptability of the improved graph structure to real-world scenarios (i.e., deployment efficiency, instruction-free, and backbone-free). In the era of LLMs, LLMs possess remarkable in-context learning capability with textual information, paving the way for efficiently obtaining improved structure. Based on this and Observation 1, we adopt a tree-oriented in-context learning approach. However, we must address **Question 2**: How can LLM be seamlessly integrated into the model architecture for efficient GSL?

**Answer 2**: In Sec. 3.2, we establish the *decoupled and training-free model design principles* by revisiting existing GSL, emphasizing efficient LLM inference over computation-intensive fine-tuning.

Based on the insights from the previous section, we propose the Large Language and Tree Assistant (LLaTA) as follows: (1) Topology-aware In-context Construction: We quantify the dynamic complexity of graph topology via structural entropy. By applying a greedy algorithm to minimize this measure, we construct a hierarchical structural encoding tree that captures topology insights focused on multi-level communities (i.e., non-leaf nodes). (2) Tree-prompted LLM Inference: The tree serves as a high-quality prompt, enabling LLMs to perform in-context learning for a deeper understanding of both topology and text. Specifically, we use LLMs to uncover textual semantic relationships within the leaf community. Based on these insights, we reallocate leaf dependencies to refine the tree structure, optimizing it within the context of the existing communities. (3) Leaf-oriented Two-step Sampling: Finally, we perform LLM-guided leaf selection to identify nodes for edge addition or removal, achieving training-free GSL. The core idea of LLaTA is to facilitate LLM in-context learning through topology-aware tree prompts, eliminating the need for costly fine-tuning. In Sec. 3.3, we provide empirical results demonstrating the effectiveness of this new GSL paradigm.

**Our contributions**. (1) *New Perspective*. We systematically review existing GSL methods and provide empirical investigations, revealing the challenges and opportunities for GSL in the era of LLMs. (2) *Innovative Approach*. We reformulate the existing GSL and propose a tree-based optimization framework with model design principles. Based on this, we introduce LLaTA, which seamlessly integrates both topology and text insights by tree-driven prompts to facilitate LLM in-context learning and generate improved graph structure, with complete theoretical support. (3) *SOTA Performance*. Extensive experiments demonstrate the superior performance of LLaTA. It outperforms recent LLM-based methods by 1.3%-2.5% in accuracy while running 2.5h-9.2h faster.

## 2 PRELIMINARIES

### 2.1 NOTATIONS AND PROBLEM FORMULATION

**Node-wise Text-Attributed Graph**. In this paper, we consider a TAG $\mathcal{G} = (\mathcal{V}, \mathcal{E})$ with $|\mathcal{V}| = n$ nodes, $|\mathcal{E}| = m$ edges. It can be described by a symmetrical adjacency matrix $\mathbf{A}(u, v)$. Each node has a feature vector of size $f$ and a one-hot label of size $c$, the feature and label matrix are represented as $\mathbf{X} \in \mathbb{R}^{n \times f}$ and $\mathbf{Y} \in \mathbb{R}^{n \times c}$. Meanwhile, $\mathcal{G}$ has node-oriented language descriptions, which are represented as $\text{t}_i \in \text{T}$ for each node and associated with its features $x_i$.

**Graph Structure Learning in TAGs**. In this paper, we aim to improve the graph structure $\mathbf{A}^\star$ for any downstream task by incorporating original topology and textual information. Given the output predictions $\hat{\mathbf{Y}}$ from the downstream backbone with improved structure, the general loss is formulated as $\mathcal{L}_{\text{task}}(\hat{\mathbf{Y}}, \mathbf{Y}) + \alpha \mathcal{L}_{\text{reg}}(\mathbf{A}^\star, \mathbf{A})$, where $\mathcal{L}_{\text{task}}$ evaluates specific task performance (e.g., node classification), and $\mathcal{L}_{\text{reg}}$ establishes the guiding principles for improved structure.

### 2.2 COMPLEXITY METRICS OF GRAPH TOPOLOGY

**Structural Entropy**. Motivated by Shannon entropy Shannon (1948), structural entropy (SE) Li & Pan (2016) is an effective measurement for quantifying the dynamic complexity of graph topology. By minimizing SE, we can reduce structure uncertainty and capture inherent patterns, thereby supporting downstream tasks in a robust and interpretable manner. In other words, this approach

uncovers meaningful structural patterns and ensures that it aligns more effectively with task-specific requirements. This has made it a pivotal tool for GNNs, gaining significant attention in recent years.

**Structural Encoding Tree**. By minimizing SE using the greedy algorithm, we can construct a structural encoding tree $\mathcal{T}$. This tree reorganizes the original graph and simulates the natural evolution of the graph through its inherent hierarchical structure. Specifically, its leaf nodes correspond to the original graph nodes, and non-leaf nodes represent multi-level communities. Low-level communities capture connected leaf nodes with high homophily, while higher levels reflect unseen structural patterns. It offers GSL a new perspective. More details can be found in Appendix A-B.

## 2.3 RELATED WORKS

**SE-based Methods**. Existing SE-based methods utilize the structural encoding tree for various graph tasks, such as node clustering Pan et al. (2021), community detection Liu et al. (2019), multi-layer coarsening Wu et al. (2022a), and embedding dimension estimation Yang et al. (2023). Recent SEGSL Zou et al. (2023) improves edge connectivity and quality with SE-based encoding tree, but it still relies on end-to-end tree-based training without incorporating LLMs.

**Traditional GSL**. These methods can be categorized into metric-based Yu et al. (2020); Zhang & Zitnik (2020); Li et al. (2022), probabilistic sampling Zheng et al. (2020); Luo et al. (2021); Liu et al. (2022a), and directly learnable methods Jin et al. (2020); Liu et al. (2022b). Specifically, they refine graph structure using well-designed measurement (e.g., homophily and connectivity), edge re-weight, random sampling, or direct topology optimization without textual information.

**LLM-based GSL**. GraphEdit Guo et al. (2024) improves node connectivity by adding neighbors through a edge predictor and refines the structure using a fine-tuned LLM. LLM4RGNN Zhang et al. (2024) leverages a fine-tuned LLM to infer node relevance, which is used to train an edge predictor. In contrast, LangGSL Su et al. (2024) integrates LLMs with graph structure learning to jointly optimize node features and graph structure, improving performance across various tasks. In Appendix C.2, we detail the advantages of LLaTA over SEGSL and LLM-based methods.

## 3 GSL IN THE ERA OF LLMs

### 3.1 GSL OPTIMIZATION OBJECTIVES REFORMULATION

As highlighted by Obs.1 in Sec. 1, existing GSL methods aim to train an effective edge predictor as the graph learner. This typically requires coupling the graph learner with a specific backbone and jointly optimizing them. Although effective, this objective becomes inefficient for LLMs due to their large number of parameters. Our investigation shows that current LLM-based GSL approaches often adopt instruction fine-tuning to mitigate full-parameter training. However, understanding complex graph structures and crafting appropriate instruction datasets can be labor-intensive, as reflected in their elaborate workflows Guo et al. (2024); Zhang et al. (2024). To address this, we reformulate GSL and introduce a tree-based optimization framework that avoids edge predictors, as illustrated in Fig. 1 (a)-(c). The core idea is to develop a language-aware tree sampler for efficient GSL. Specifically, (a) Tree Construction first reorganizes the original graph to assist LLMs in understanding the topology. Based on this, (b) Tree Optimization leverages textual information and LLM to perform tree optimization. Finally, (c) Improved Structure is generated by a language-aware tree sampler. Please refer to Sec.3.2 and Sec.4 for implementation of the above framework in LLaTA.

### 3.2 MODEL ARCHITECTURE REVIEW

As outlined by Obs. 2 in Sec. 1, integrating LLMs in GSL remains a challenge. To establish model design principles for tree optimization, we revisit existing GSL paradigms. The **Coupled Paradigm** tightly integrates the graph learner and backbone for task-specific learning. However, it faces limitations such as unstable performance due to restricted generalizability and module dependent, and inflexible deployment as switching tasks or backbones requires retraining. In contrast, the **Decoupled Paradigm** trains the graph learner and backbone independently, offering better compatibility with LLMs for GSL. Although recent LLM-based GSL methods adopt this approach, they still face challenges such as indirect integration of topology and text, reliance on instruction datasets, and the complexity of fine-tuning. For a more efficient and reliable approach, we recommend a fully decoupled paradigm, reducing fine-tuning reliance and focusing on high-quality in-context prompts. Detailed discussion about the GSL paradigms is provided in the Appendix C.1.

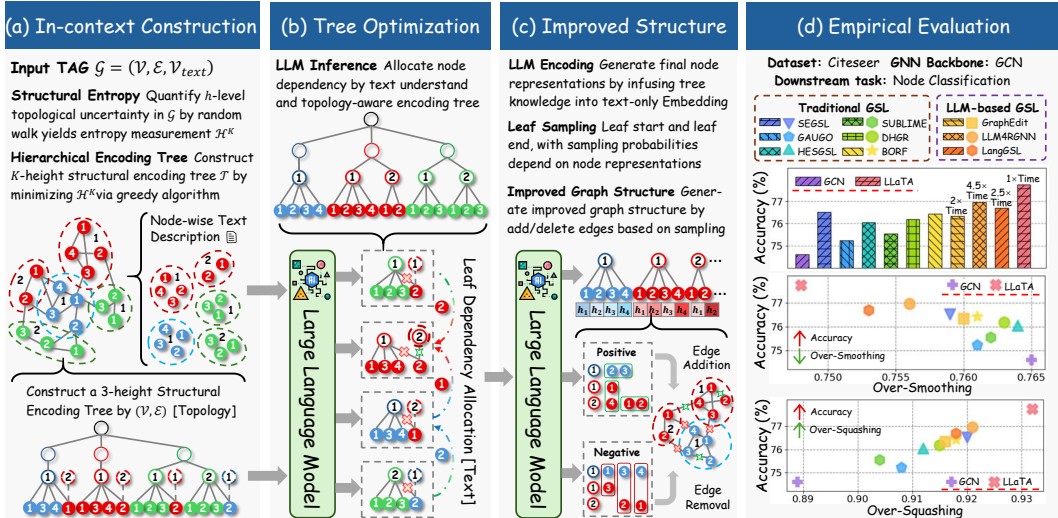

Figure 1: The overview of our proposed tree-based GSL optimization pipeline and empirical results.

To achieve this, we leverage the in-context learning ability of LLMs to jointly understand graph topology and node text for reliable inference. Specifically, **(a) Tree Construction**: we first build a structural encoding tree to generate topology-aware in-context prompts, enhancing the LLM's understanding of graph structure. Then, **(b) Tree Optimization**: based on these prompts, we use LLMs to infer semantic relationships among nodes from their textual descriptions, and reassign leaf dependencies to optimize the tree structure based on the community structure. The resulting hierarchical tree integrates both structural and textual information, forming a strong basis for GSL. Finally, **(c) Improved Structure**: we apply a leaf-oriented two-step sampling to select the current node and a candidate set for edge modification, yielding an improved graph for any downstream scenario. For deeper insights into the leaf-oriented sampler within GSL, please refer to Appendix B.

### 3.3 EMPIRICAL INVESTIGATION

To demonstrate the effectiveness of our proposal in Sec. 3.1-3.2, we present empirical results in Fig. 1 (d). Due to space constraints, detailed experimental settings and analysis are provided in Appendix D. In our reports, over-smoothing ($\downarrow$) reflects the distinguishability of node embeddings, while over-squashing ($\uparrow$) indicates the ability of nodes to perceive distant ones. These metrics are commonly used in recent GSL studies to quantify the quality of improved structure, as they not only directly impact downstream accuracy ($\uparrow$) but also evaluate method robustness. They demonstrate that LLM-based GSL methods generate higher-quality structures than traditional ones, underscoring the necessity of LLMs. Furthermore, LLaTA achieves the best performance, validating the effectiveness of our proposed new GSL paradigm.

## 4 OUR METHOD

We instantiate the three components outlined in Fig. 1 and propose LLaTA, shown in Fig. 2. The correspondences are: (a)-(1) Topology-aware In-context, (b)-(2) Tree-prompted LLM Inference and (c)-(3) Leaf-oriented Two-step Sampling.

### 4.1 TOPOLOGY-AWARE IN-CONTEXT CONSTRUCTION

**Motivation**. To address the potential performance decline caused by the absence of fine-tuning and obtain reliable inference, we enable efficient in-context learning with high-quality prompts. Based on Sec. 2.2, structural entropy is a promising strategy for constructing these prompts, as it offers an interpretable theory for capturing both local and global topology insights.

To establish a hierarchical encoding tree that simulates the natural evolution of graph topology, we constrain the height of this tree to $K$ and minimize $K$-dimensional SE proposed by Li & Pan (2016); Zou et al. (2023). The optimization objective definition:

$$\mathcal{T}^{\star} = \underset{\mathcal{T}}{\arg\min} \, \mathcal{H}^{\mathcal{T}}(\mathcal{G}), \quad \mathcal{H}^{\mathcal{T}}(\mathcal{G}) = \sum_{\phi \in \mathcal{T}, \phi \neq \lambda} \mathcal{H}^{\mathcal{T}}(\mathcal{G}, \phi) = -\sum_{\phi \in \mathcal{T}, \phi \neq \lambda} \frac{g_\phi}{\text{vol}(\mathcal{G})} \log \frac{\text{vol}(\phi)}{\text{vol}(\phi^+)}, \tag{1}$$

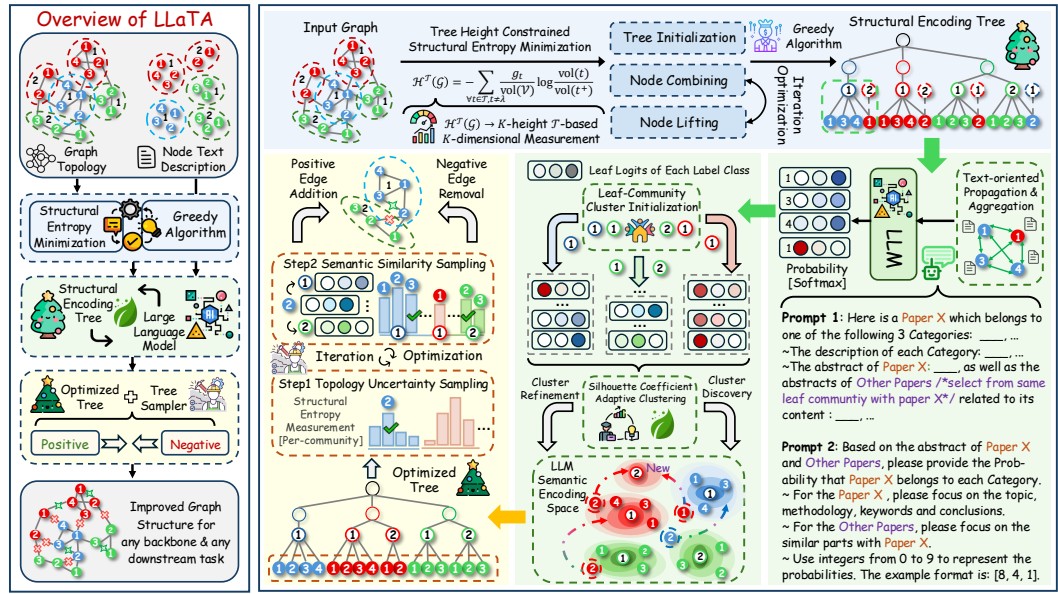

Figure 2: (Left) The overview of LLaTA; (Right) The detailed pipeline of LLaTA, which includes: topology-aware tree prompts, reliable LLM inference and language-aware tree sampler.

where $\text{vol}(\mathcal{G})$ denotes the total degree of $\mathcal{G}$, $g_\phi$ is the sum of edges crossing the tree node, $\text{vol}(\phi)$ is the sum of the degrees of nodes in $\phi$, $\phi^+$ is the parent of $\phi$, $\lambda$ is the root node, $\mathcal{T}^\star$ denotes the optimized tree that minimizes its structural entropy $\mathcal{H}^\mathcal{T}(\mathcal{G})$. We employ a greedy algorithm to minimize the structural entropy $\mathcal{H}^\mathcal{T}(\mathcal{G})$ and construct a hierarchical encoding tree $\mathcal{T}^\star$ with a predefined height $K$. The algorithm starts by initializing a tree of height 1, where each graph node is a leaf under a common root. It then iteratively combines node pairs to reduce $\mathcal{H}^\mathcal{T}(\mathcal{G})$, followed by node lifting operations to further optimize the structure. The node lifting process continues until the tree height is reduced to $K$. The full algorithm and the operations involved is detailed in Appendix E.1.

**Theorem 1** (Topological Information Capturing Properties of Encoding Tree). *Given an encoding tree $\mathcal{T}$ and a non-leaf node $\phi \in \mathcal{T}$, the error of topological information $\varepsilon^h(\phi)$ in community $\mathcal{C}_\phi$ is upper bounded by: $\frac{g_\phi}{2m} \log_2 \frac{\text{vol}(\phi^+)}{g_\phi}$, and $\varepsilon^h(\phi)$ gradually decreases as the community level descends.*

**Theorem 2** (Implicit Global Constraints in Low-Level Communities). *In a structural encoding tree $\mathcal{T}$, each low-level community $\mathcal{C}^\ell$ captures localized topology but implicitly retains global structure constraints due to the hierarchical, random-walk-based formulation of structural entropy.*

Drawing on Theorems 1 and 2, we infer that low-level communities in the encoding tree provide a more faithful representation of the graph's topology, as they simultaneously preserve local connectivity and global structural constraints. Accordingly, for each leaf node $\alpha$, we assign its associated low-level community $\mathcal{C}^\ell_{\alpha^+}$ as a topological context to enhance the accuracy of downstream inference by the LLMs. The proof process of Theorems 1 and 2 is provided in Appendix F.1-F.2.

## 4.2 TREE-PROMPTED LLM INFERENCE

**Motivation**: Based on the structural encoding tree $\mathcal{T}$, we further integrate node language descriptions to enable comprehensive in-context learning with LLMs, fostering a better understanding of both topology and textual information in the original graph. This allows LLMs to make reliable inferences without fine-tuning. These inferences are then used to optimize the encoding tree, enhancing homophily within low-level communities and laying a strong foundation for subsequent GSL.

**Reception-aware Leaf Augmentation**. To construct effective in-context prompts for LLMs, the quality of node-specific text is essential. Thus, we perform tree-guided text propagation and aggregation within each low-level community $\mathcal{C}^\ell$ to achieve leaf augmentation for $\epsilon$-guaranteed $t^\star$. The key idea is that nodes in low-level communities typically show high connectivity and homophily, enabling efficient text sharing. Using this property, we filter and combine relevant texts for each leaf

node based on a similarity threshold $\epsilon$, helping to reduce noise. This process is formally defined as:

$$t_\alpha^\star = \text{Concat}\left(\{t_\alpha\} \cup \{t_j \mid j \in \text{Top}_k\left(\{w_{\alpha\beta} \mid \beta \in \mathcal{C}_{\alpha+}^\ell\}\right), w_{\alpha\beta} \geq \epsilon\}\right), \tag{2}$$

$$w_{\alpha\beta} = \frac{\exp\left(\text{sim}(x_\alpha, x_\beta)\right)}{\sum\limits_{\gamma \in \mathcal{C}_{\alpha+}^\ell} \exp\left(\text{sim}(x_\alpha, x_\gamma)\right)}, \quad \text{sim}(x_\alpha, x_\beta) = \frac{\sum_{i=1}^{f\text{-dim}} x_\alpha^i \cdot x_\beta^i}{\sqrt{\sum_{i=1}^{f\text{-dim}} (x_\alpha^i)^2 \sum_{i=1}^{f\text{-dim}} (x_\beta^i)^2}}. \tag{3}$$

**Community of Thought**. Building on the enhanced leaf nodes, we propose a *Community of Thought* (CoT) prompt mechanism, where each node is represented together with its community-aware descriptions and related neighbors. This design enables LLMs to jointly capture topological and semantic information, thereby supporting more reliable inference. Through this mechanism, the LLM can (i) infer the probability distribution over label classes for each node, and (ii) generate high-quality embeddings within its semantic space. These embeddings are then employed for fine-grained optimization of the structural encoding tree. In summary, CoT facilitates efficient in-context learning for LLMs by leveraging the structure-aware encoding tree. Formally, given a target leaf node $\alpha$, the LLM inference with CoT can be expressed as:

$$\mathbf{z}_\alpha = \Psi(\text{LLM-CoT}(\mathcal{D}_{\text{task}}, \mathcal{D}_{\text{topo}}, \mathcal{D}_{\text{sema}}, t_\alpha^\star)), \quad \Psi : \text{ANS}_\alpha \mapsto \mathbb{R}^{N_{\text{class}}}, \tag{4}$$

$$y_\alpha^{cls} \sim \text{Categorical}(p_\theta(y \mid \mathbf{z}_\alpha)), \quad p_\theta(y = i \mid \mathbf{z}_\alpha) = \frac{\exp(z_{\alpha,i})}{\sum_{j=1}^{N_{\text{class}}} \exp(z_{\alpha,j})}, \quad i = 1, \ldots, N_{\text{class}}, \tag{5}$$

where $\Phi(\cdot)$ is the extraction function that maps the LLM's textual answer $\text{ANS}_\alpha$ into a numerical logit vector $\mathbf{z}_\alpha \in \mathbb{R}^{N_{\text{class}}}$, $y_\alpha^{cls} \in \Delta^{N_{\text{class}}-1}$ denotes the soft label probability simplex for node $\alpha$. Here, $\mathcal{D}_{\text{task}}$ specifies the task, while $\mathcal{D}_{\text{topo}}$ and $\mathcal{D}_{\text{sema}}$ encode topological and semantic context, respectively. An illustration of prompt construction is provided in Fig. 2.

**Theorem 3** (Error Bound Between Soft labels and True Labels)**.** *Given two leaf nodes $\alpha$ and $\beta$ in $\mathcal{T}$, the error between soft label similarity and true label similarity is bounded by:*

$\varepsilon^y(\alpha\beta) = |\text{sim}(y_\alpha^{cls}, y_\beta^{cls}) - \text{sim}(y_\alpha, y_\beta)| \leq \delta \cdot (1 - \epsilon)$, *where $\delta$ is a constant that depends on the LLM's in-context learning ability and $y_\alpha$ is the true label of $\alpha$. (The proof is detailed in Appendix F.3.)*

Based on Theorem 3, we have that the similarity of soft labels $y^{cls}$ can approximate the similarity of ground-truth labels $y$ with controllable bias $\varepsilon^y(\alpha\beta)$. This provides robust theoretical guarantee and support for the subsequent **Leaf Dependency Allocation** and **Semantic Similarity Sampling**.

**Leaf Dependency Allocation.** While the structural encoding tree captures graph topology, effective GSL also requires incorporating text-driven node attributes. To this end, we optimize low-level communities by reallocating leaf dependencies under the guidance of LLM-derived soft labels. Specifically, we (i) initialize clusters from the original structural encoding tree, (ii) reassign minority-label nodes to align with the dominant class, and (iii) adaptively refine cluster granularity using the silhouette coefficient to form communities with stronger homophily. This yields the optimized structural encoding tree $\mathcal{T}^\star$, which serves as the foundation for tree sampling (see Fig. 2). The procedure can be compactly expressed as:

$$\mathcal{P}^\star = \arg\max_{\mathcal{P}} \left\{ \sum_{\mathcal{C} \in \mathcal{P}} \max_c \sum_{v \in \mathcal{C}} y_{v,c}^{cls} + \beta \sum_{\mathcal{C} \in \mathcal{P}} \text{Sil}(\mathcal{C}; \mathbf{Y}^{cls}) \right\}, \quad \mathcal{T}^\star = \text{UpdateTree}(\mathcal{T}, \mathcal{P}^\star). \tag{6}$$

Here, $\mathcal{P}$ denotes a partition of the leaf nodes into communities, $y_{v,c}^{cls}$ is the probability of node $v$ belonging to class $c$, and $\text{Sil}(\mathcal{C}; \mathbf{Y}^{cls})$ is the silhouette coefficient evaluating the homophily of community $\mathcal{C}$. The optimized partition $\mathcal{P}^\star$ is then used to update the structural encoding tree $\mathcal{T}$, yielding the refined tree $\mathcal{T}^\star$. The adaptive clustering algorithm is detailed in Appendix E.2.

### 4.3 LEAF-ORIENTED TWO-STEP SAMPLING

**Motivation**. Although structural optimization can be applied to all leaf nodes, we adopt a two-step sampling technology to balance running efficiency and practical performance. Specifically, we first identify a limited set of nodes requiring optimization from a topological perspective. Subsequently, we select a candidate set of nodes closely related to these nodes from a text semantic perspective. Finally, through edge addition or removal, we facilitate training-free GSL. The detailed algorithm of the two-step sampling can ben found in Appendix E.3.

**Theorem 4** (High-Entropy Nodes Require Supervision). *In a structural encoding tree $\mathcal{T}$ constructed via entropy minimization, nodes with higher structural entropy $\mathcal{H}^{\mathcal{T}}(\mathcal{G}, \alpha)$ indicate: higher topological uncertainty within their local structural context. (The proof is detailed in Appendix F.4.)*

We perform two-step sampling under the guidance of the following two probability functions:

$$P_{topo}(\alpha) = \frac{\exp(\mathcal{H}^{\mathcal{T}}(\mathcal{G}, \alpha))}{\sum\limits_{\gamma \in \mathcal{C}_{\alpha+}^{\ell}} \exp(H^{\mathcal{T}}(\mathcal{G}, \gamma))}, \quad P_{sema}^{\alpha}(\beta) = \frac{\exp(\text{sim}(y_{\beta}^{cls}, y_{\alpha}^{cls}))}{\sum\limits_{\gamma \in \mathcal{C}_{\alpha+}^{\ell}, \gamma \neq \alpha} \exp(\text{sim}(y_{\gamma}^{cls}, y_{\alpha}^{cls}))}. \tag{7}$$

**Topology Uncertainty Sampling.** The core of GSL is to eliminate structural noise and enhance data utility. Notably, this noise usually appears only in certain substructures rather than the entire graph. According to Theorem 4, nodes with higher structural entropy are more likely to lie in topologically uncertain regions and therefore benefit more from structure refinement. Based on $P_{topo}(\alpha)$ in Eq. 7, we prioritize nodes for sampling within each leaf community $\mathcal{C}^{\ell}$ according to their structural entropy, improving efficiency by focusing on nodes most in need of optimization.

**Semantic Similarity Sampling.** After selecting a node $\alpha$, we further determine candidate neighbors using $P_{sema}^{\alpha}(\beta)$ in Eq. 7, which measures LLM-empowered semantic similarity between soft labels $y^{cls}$. A $\theta$-sized candidate set is formed from the remaining leaf nodes, and edges are updated accordingly: for edge addition, nodes are ranked by descending similarity; for edge removal, by ascending similarity. This ensures that edge modifications are guided by both semantic consistency and structural refinement.

## 5 EXPERIMENT

In this section, we conduct a wide range of experiments and aim to answer: **Q1: Effectiveness**. Compared with other state-of-the-art GSL methods, can LLaTA achieve better performance? **Q2: Interpretability**. If LLaTA is effective, what contributes to its outstanding performance? **Q3: Robustness**. How does LLaTA perform when deployed in real-world complex scenarios? **Q4: Efficiency**. How efficient is LLaTA compared to other competitive GSL methods?

### 5.1 EXPERIMENT SETUP

We evaluate LLaTA and 14 GSL baselines on 11 widely adopted TAG datasets across multiple domains. The details on these datasets and baselines can be found in Appendix G-H. Due to space limitations, more experimental setup and hyperparameter details are provided in Appendix I.

### 5.2 PERFORMANCE COMPARISON

**Node Classification**. To answer **Q1**, we first present the node classification performance results in Table 1, where LLaTA consistently outperforms other baselines. For instance, LLaTA outperforms the second-best method by 1.18%, 1.02%, and 1.07% on the Cora, WikiCS, and Instagram datasets, respectively. Moreover, LLaTA demonstrates significant gains in complex History and Photo, outperforming the second-best methods by 2.45% and 2.78%. Notably, on the Child dataset, LLaTA achieves an impressive improvement of 9.87%, highlighting its strong capability in handling heterophily. These results underscore LLaTA's effectiveness in achieving superior performance.

**Node Clustering**. To further answer **Q1**, we conducted a comprehensive comparison of LLaTA with GSL methods that have demonstrated strong performance in node classification, extending to node clustering. As shown in Table 2, LLaTA consistently outperforms competing methods across all four datasets, significantly enhancing the effectiveness of GNNs in unsupervised tasks. These results confirm that LLaTA is applicable to diverse tasks, demonstrating its generalization and adaptability.

### 5.3 ABLATION STUDY

To answer **Q2**, we conduct an ablation study shown in Table 3 and provide a case study in Appendix J, which provides a detailed analysis of our method's pipeline by visualizing the data flow. Meanwhile, we provide LLM backbone analysis in Appendix K. In the ablation study, we use TO to denote tree optimization, while $TO_{LLM}$ and $SAM_{LLM}$ denote the tree optimization and sampling processes with LLM. For the ablation setup, the LLM inference results are replaced by the initial features. Additionally, $LLM_{NE}$ and $LLM_{RW}$ represent LLM inference with topology-aware tree in-context information derived from 1-hop neighbors and random walk sequences, rather than the tree, respectively.

Table 1: Node classification accuracy(%) on 11 TAG datasets. Highlighted are the top **first**, **second**, and **third** accuracy. "OOM" denotes out of memory and "OOT" denotes out of time.

| Method | Cora | Citeseer | Pubmed | WikiCS | Instagram | Reddit | Ratings | Child | History | Photo | ArXiv |
|---|---|---|---|---|---|---|---|---|---|---|---|
| IDGL | $86.81_{\pm0.31}$ | $75.39_{\pm0.27}$ | $87.25_{\pm0.22}$ | $79.78_{\pm0.14}$ | $63.68_{\pm0.09}$ | $65.03_{\pm0.13}$ | $43.29_{\pm0.29}$ | $45.33_{\pm0.26}$ | $80.46_{\pm0.31}$ | $82.33_{\pm0.33}$ | OOM |
| SLAPS | $79.89_{\pm0.50}$ | $73.51_{\pm0.64}$ | $86.41_{\pm0.49}$ | $72.50_{\pm0.27}$ | $60.91_{\pm0.13}$ | OOM | $40.76_{\pm0.23}$ | $47.46_{\pm0.21}$ | OOM | OOM | OOM |
| GAUGO | $85.24_{\pm0.27}$ | $75.24_{\pm0.43}$ | $87.12_{\pm0.38}$ | $70.74_{\pm0.29}$ | $63.34_{\pm0.21}$ | OOM | $40.19_{\pm0.24}$ | $47.27_{\pm0.19}$ | OOM | OOM | OOM |
| HESGSL | $85.91_{\pm0.45}$ | $76.03_{\pm0.31}$ | $87.17_{\pm0.27}$ | $73.29_{\pm0.22}$ | $63.83_{\pm0.16}$ | OOM | $38.71_{\pm0.42}$ | $47.81_{\pm0.25}$ | OOM | OOM | OOM |
| SEGSL | $86.90_{\pm0.54}$ | $76.52_{\pm0.20}$ | $87.42_{\pm0.38}$ | $79.68_{\pm0.29}$ | $65.26_{\pm0.21}$ | $64.87_{\pm0.26}$ | $43.20_{\pm0.47}$ | $51.80_{\pm0.39}$ | OOT | OOT | OOT |
| ProGNN | $84.18_{\pm0.23}$ | $75.08_{\pm0.21}$ | $86.89_{\pm0.14}$ | $71.90_{\pm0.16}$ | $64.45_{\pm0.19}$ | OOM | OOM | OOM | OOM | OOM | OOM |
| SUBLIME | $84.81_{\pm0.33}$ | $75.55_{\pm0.47}$ | $87.63_{\pm0.70}$ | $78.06_{\pm0.38}$ | $64.78_{\pm0.24}$ | $58.82_{\pm0.21}$ | $39.90_{\pm0.17}$ | $49.54_{\pm0.16}$ | OOM | OOM | OOM |
| STABLE | $84.21_{\pm0.43}$ | $75.34_{\pm0.60}$ | $87.27_{\pm0.39}$ | $76.93_{\pm0.33}$ | $65.66_{\pm0.18}$ | $59.78_{\pm0.24}$ | $40.55_{\pm0.10}$ | $50.90_{\pm0.09}$ | OOM | OOM | OOM |
| CoGSL | $86.16_{\pm0.45}$ | $75.45_{\pm0.36}$ | $86.52_{\pm0.49}$ | OOM | OOM | OOM | OOM | OOM | OOM | OOM | OOM |
| BORF | $87.08_{\pm0.18}$ | $76.43_{\pm0.17}$ | $87.45_{\pm0.11}$ | $80.56_{\pm0.08}$ | $64.07_{\pm0.06}$ | $63.07_{\pm0.09}$ | $42.72_{\pm0.60}$ | $50.92_{\pm0.54}$ | $82.83_{\pm0.14}$ | $82.63_{\pm0.15}$ | $73.08_{\pm0.33}$ |
| DHGR | $86.72_{\pm0.42}$ | $76.18_{\pm0.14}$ | $86.76_{\pm0.33}$ | $79.25_{\pm0.17}$ | $64.26_{\pm0.26}$ | $66.41_{\pm0.29}$ | $43.33_{\pm0.36}$ | $52.41_{\pm0.40}$ | $82.61_{\pm0.16}$ | $81.92_{\pm0.18}$ | $73.47_{\pm0.28}$ |
| GraphEdit | $86.26_{\pm1.06}$ | $76.33_{\pm1.12}$ | $87.14_{\pm0.29}$ | $79.92_{\pm0.63}$ | $64.23_{\pm1.01}$ | OOT | $42.53_{\pm0.97}$ | OOT | OOT | OOT | OOT |
| LLM4RGNN | $86.73_{\pm0.73}$ | $76.96_{\pm0.58}$ | $87.37_{\pm0.21}$ | OOT | $64.85_{\pm0.66}$ | OOT | OOT | OOT | OOT | OOT | OOT |
| LangGSL | $87.65_{\pm0.67}$ | $76.69_{\pm0.73}$ | $87.24_{\pm1.21}$ | $80.26_{\pm0.45}$ | $66.25_{\pm0.39}$ | $66.87_{\pm0.12}$ | $44.01_{\pm1.37}$ | $49.66_{\pm1.05}$ | OOT | OOT | OOT |
| LLaTA (Ours) | $88.26_{\pm0.26}$ | $78.21_{\pm0.18}$ | $88.39_{\pm0.23}$ | $81.58_{\pm0.20}$ | $66.73_{\pm0.13}$ | $67.60_{\pm0.19}$ | $44.47_{\pm0.08}$ | $62.28_{\pm0.56}$ | $85.28_{\pm0.24}$ | $85.41_{\pm0.24}$ | $75.39_{\pm0.41}$ |

Table 2: Comparison of node clustering.

| Method | Cora | Citeseer | Pubmed | Instagram |
|---|---|---|---|---|
| SEGSL | $67.70_{\pm0.17}$ | $71.15_{\pm0.12}$ | $79.05_{\pm0.04}$ | $64.63_{\pm0.01}$ |
| BORF | $67.54_{\pm0.16}$ | $70.62_{\pm0.10}$ | $79.46_{\pm0.03}$ | $64.49_{\pm0.02}$ |
| DHGR | $66.85_{\pm0.63}$ | $70.14_{\pm0.21}$ | $78.64_{\pm0.15}$ | $64.52_{\pm0.06}$ |
| LLM4RGNN | $67.63_{\pm0.11}$ | $71.72_{\pm0.13}$ | $78.90_{\pm0.06}$ | OOT |
| LangGSL | $67.82_{\pm0.16}$ | $71.38_{\pm0.14}$ | $79.95_{\pm0.07}$ | $65.03_{\pm0.01}$ |
| LLaTA | $68.39_{\pm0.14}$ | $72.18_{\pm0.15}$ | $80.25_{\pm0.06}$ | $65.37_{\pm0.02}$ |

Table 3: Ablation study result of LLaTA.

| Component | Pubmed | WikiCS | Instagram | Ratings |
|---|---|---|---|---|
| w/o TO | $85.34_{\pm0.35}$ | $78.63_{\pm0.28}$ | $64.06_{\pm0.23}$ | $42.05_{\pm0.14}$ |
| w/o TO$_{LLM}$ | $86.81_{\pm0.26}$ | $80.03_{\pm0.22}$ | $64.94_{\pm0.15}$ | $42.87_{\pm0.11}$ |
| w/o SAM$_{LLM}$ | $86.68_{\pm0.30}$ | $80.23_{\pm0.25}$ | $64.81_{\pm0.18}$ | $42.72_{\pm0.13}$ |
| w/ LLM$_{NE}$ | $87.04_{\pm0.21}$ | $80.09_{\pm0.20}$ | $65.13_{\pm0.14}$ | $41.96_{\pm0.09}$ |
| w/ LLM$_{RW}$ | $87.23_{\pm0.21}$ | $80.46_{\pm0.19}$ | $65.57_{\pm0.15}$ | $42.40_{\pm0.11}$ |
| LLaTA | $88.39_{\pm0.23}$ | $81.58_{\pm0.20}$ | $66.73_{\pm0.13}$ | $44.47_{\pm0.08}$ |

From the ablation study, we obtain the following key conclusions: (1) Removing TO leads to a significant performance decline, indicating that the tree plays a crucial role in LLM-based GSL. (2) Replacing LLM inference results with initial node features to guide TO and SAM leads to a performance decline. This highlights the critical role of LLM-generated contextual information, as initial node features alone are inadequate for capturing complex semantic relationships. (3) Utilizing 1-hop neighbors and random walk sequences to provide topology-aware in-context information also results in performance degradation. This highlights the necessity of the SE-based hierarchical structural encoding tree. Meanwhile, SE provides valuable guidance for node sampling, a capability that cannot be effectively achieved solely with 1-hop neighbors or random walk sequences.

### 5.4 ROBUSTNESS ANALYSIS

To answer **Q3**, we conduct a thorough analysis of the LLaTA's robustness from the following aspects:

**Downstream Backbones**. According to Table 4, LLaTA consistently improves performance of the backbones across all four datasets, with gains ranging from 0.56% to 10.18%. These enhancements are observed not only on conventional GNN backbones (GCN, GAT, GraphSAGE) but also on LLM-GNN backbones (GLEM, ENGINE), highlighting the effectiveness and applicability of our structure optimization across diverse downstream architectures.

**Real-world Scenarios**. To evaluate the robustness of LLaTA against sparsity and noise scenarios in practical applications, we randomly remove or add edges to the original graph structure. As illustrated in Fig. 3, LLaTA demonstrates strong predictive performance in both scenarios. In the sparse setting, LLaTA achieves optimal performance across most perturbation levels. In the noisy scenario, LLaTA performs competitively, with only LLM4RGNN surpassing it, as the latter is specifically designed to handle graph adversarial attack settings. Notably, as the edge addition rate increases, the performance of LLaTA gradually deteriorates, which can be attributed to the irreversible negative impact of such perturbations on the tree initialization. Further experimental results are provided in the Appendix L.

**Hyperparameter**. We conducted a comprehensive analysis of the four key hyperparameters of LLaTA: $K, \epsilon, \theta$, and $r$. The experimental results for $K, \theta$, and $r$ are presented in Fig. 4, while the more experimental analysis and the detailed selection guidance of the four hyperparameters are provided in Appendix M. In Fig. 4 (left), the optimal tree height $K$ differs across datasets, reflecting varying graph complexity. However, when $K$ reaches 6, performance drops markedly for all datasets, suggesting that excessive tree depth may cause overfitting or loss of structural information. In Fig. 4 (right), we analyze LLaTA's sensitivity to the candidate set size $\theta$ in semantic similarity sampling and the two-step sampling frequency $r$ on the History dataset. The results indicate that an $r$ value between 3 and 10 yields higher accuracy, corresponding to an appropriately improved graph density. In contrast, LLaTA is less sensitive to the choice of $\theta$, achieving stable performance within the range of $\theta \in [5, 15]$. A smaller $\theta$ may result in insufficient semantic similarity sampling, whereas a larger $\theta$ could introduce noise into the candidate set, negatively impacting performance.

Table 4: Improvements of LLaTA over different GNN/LLM-GNN backbones.

| Backbone | Reddit | Child | History | Photo |
|---|---|---|---|---|
| LLaTA$_{GCN}$ | $67.60_{\pm 0.19}$ | $62.28_{\pm 0.56}$ | $85.28_{\pm 0.24}$ | $85.41_{\pm 0.24}$ |
| LLaTA$_{GAT}$ | $63.80_{\pm 0.23}$ | $62.65_{\pm 0.71}$ | $85.30_{\pm 0.29}$ | $85.38_{\pm 0.33}$ |
| LLaTA$_{SAGE}$ | $60.73_{\pm 0.30}$ | $63.53_{\pm 0.99}$ | $84.67_{\pm 0.45}$ | $82.20_{\pm 0.51}$ |
| LLaTA$_{GLEM}$ | $68.37_{\pm 0.08}$ | $64.49_{\pm 0.35}$ | $87.52_{\pm 0.29}$ | $86.99_{\pm 0.26}$ |
| LLaTA$_{ENGINE}$ | $\mathbf{69.57_{\pm 0.31}}$ | $\mathbf{64.75_{\pm 0.52}}$ | $\mathbf{88.43_{\pm 0.21}}$ | $\mathbf{87.59_{\pm 0.22}}$ |
| Improvement | $\uparrow 1.79{\sim}2.76$ | $\uparrow 8.27{\sim}10.18$ | $\uparrow 0.56{\sim}3.07$ | $\uparrow 1.14{\sim}2.98$ |

Table 5: Time complexity analysis of LLaTA and other GSL method.

| Method | Time Complexity |
|---|---|
| SEGSL | $\mathcal{O}(n^2 + nlog^2n + n)$ |
| GraphEdit | $\mathcal{O}(pt_{\text{LLM-r}} + qd^2 + n^2 + (m + nk)t_{\text{LLM-i}})$ |
| LLM4RGNN | $\mathcal{O}(pt_{\text{LLM-r}} + mt_{\text{LLM-i}} + qd^2 + n^2)$ |
| LangGSL | $\mathcal{O}(nt_{\text{LLM-i}} + nt_{\text{LM-e}} + n^2 + nd^2 + n^2d)$ |
| LLaTA | $\mathcal{O}(nlog^2n + n|\mathcal{C}|^2 + nt_{\text{LLM-i}} + n)$ |

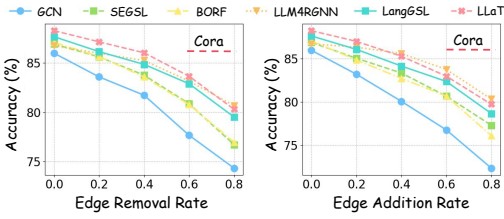
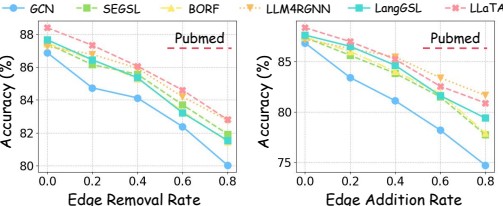

Figure 3: Node classification performance on the Cora and Pubmed dataset under real-world scenarios (Sparsity [Edge Removal] and Noise [Edge Addition]).

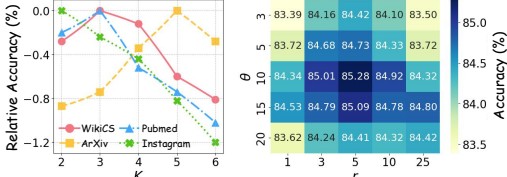
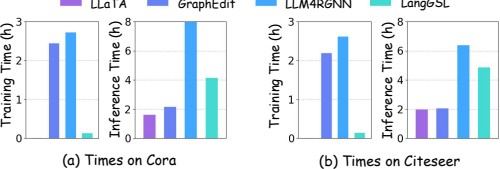

Figure 4: Hyperparameter analysis of $K$, $\theta$ and $r$. $\theta$ and $r$ are analyzed on the History dataset.

Figure 5: Comparison of training and inference time for LLM-based GSL method.

## 5.5 EFFICIENCY ANALYSIS

To answer **Q4**, we conduct a theoretical analysis of the algorithm time complexity of LLaTA and empirically demonstrate its efficiency in integrating LLMs with GSL.

In Table 5, we analyze the theoretical time complexity of LLaTA and existing LLM-based GSL methods. Where $q$ and $p$ is the number of samples for fine-tuning and edge predictor training, $d$ denotes the feature dimension and $t_{\text{LM-e}}$ represents the LM encoding time. $t_{\text{LLM-r}}$ and $t_{\text{LLM-i}}$ denote the fine-tuning and inference time of LLM, respectively. The magnitude of $t_{\text{LLM-i}}$ should be adaptively determined according to the specific requirements of different inference tasks. For LLaTA, The time complexity can be divided into three parts: (a) The complexity for tree construction in Sec. 4.1 is $\mathcal{O}(nlog^2n)$, (b) The complexity of tree-prompted LLM inference in Sec. 4.2 is $\mathcal{O}(n|\mathcal{C}|^2 + nt_{\text{LLM-i}})$, (c) The complexity of the two-step sampling in Sec. 4.3 is $\mathcal{O}(\theta n)$. Complete runtime results and scalability analysis of LLaTA across all datasets are provided Appendix N.

To practically evaluate the efficiency of LLaTA, we compare its training and inference time with existing LLM-based GSL methods. The training time includes the fine-tuning process of the LLM and the training of the edge predictor, while the inference time encompasses LLM inference and other GSL modules. As shown in Fig. 5, the experimental results demonstrate that our method requires significantly less inference time, particularly when compared to LLM4RGNN. Furthermore, our approach eliminates the need for any training during the structure learning phase, resulting in zero training time. This highlights the efficiency of our method in seamlessly integrating LLMs with GSL.

## 6 CONCLUSION

In this paper, we first introduce a novel optimization framework for GSL by rethinking its integration with LLMs, addressing the challenges of incorporating textual information. Subsequently, We propose LLaTA as an instantiation of this novel framework, which leverages a structural encoding tree to achieve efficient LLM in-context learning. This enables the LLM to comprehensively understand both topology and text insights from the original graph, ultimately relying on reliable inference to obtain improved graph structure. To this end, LLaTA eliminates the need for costly fine-tuning and achieves SOTA performance. Inspired by the noise experiments, a promising direction is developing more robust tree optimization algorithms, laying a solid foundation for LLM-empowered GSL.

## REPRODUCIBILITY STATEMENT

To ensure reproducibility, we provide detailed LLaTA's model architecture design, theoretical analysis, and algorithm pseudocode in Sec. 3.2, Sec. 4, Appendix F and Appendix E. The detailed information of the dataset used in the experiment can be found in Appendix G, the introduction of the baselines used in the experiment can be found in Appendix H, and detailed experimental and hyperparameter settings can be found in the Appendix I and Appendix M. Codes and other necessary materials are provided in our supplementary materials.

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

## A  ENCODING TREE AND STRUCTURAL ENTROPY

In this section, we provide a detailed exposition of the formal definitions of the encoding tree and related structural entropy. Based on this, we present a visualized toy example in Fig. 6 to intuitively illustrate the practical significance of leaf nodes, low-level communities, and higher-level communities, offering readers a clearer conceptual understanding.

**Definition 1.** *Encoding Tree.* *The language-aware encoding tree $\mathcal{T}$ of a node-wise TAG $\mathcal{G} = (\mathcal{V}, \mathcal{E}, \mathrm{T})$ satisfies the following properties:*

*(1) The root node $\lambda$ has a community $\mathcal{C}_\lambda = \mathcal{V}$, where $\mathcal{V}$ represents the set of all vertices in $\mathcal{G}$.*

*(2) Each non-root node $\alpha$ corresponds to a community $\mathcal{C}_\alpha \subset \mathcal{V}$. If $\alpha$ is a leaf node, $\mathcal{C}_\alpha$ is a singleton with one vertex from $\mathcal{V}$.*

*(3) For each leaf node $\alpha$ with $\mathcal{C}_\alpha = v_i$, $x_\alpha = x_i$ is the initial embedding of $\alpha$, and $\mathrm{t}_\alpha = \mathrm{t}_i$ is the raw text of $\alpha$.*

*(4) For each non-root node $\alpha$, the parent node corresponding to its community which it belongs to is denoted as $\alpha^+$.*

*(5) Each non-leaf node $\alpha$, its $i$-th child node is denoted as $\alpha^{\langle i \rangle}$, where the children are ordered from left to right as $i$ increases. If $\alpha$ is not $\lambda$, it can be considered as a community.*

*(6) For each non-leaf node $\alpha$, assuming it has $m$ children, the communities of its children $\mathcal{C}_{\alpha^{\langle i \rangle}}$ together form the community of $\mathcal{C}_\alpha$, so that $\mathcal{C}_\alpha = \bigcup_{i=1}^m \mathcal{C}_{\alpha^{\langle i \rangle}}$ and $\bigcap_{i=1}^m \mathcal{C}_{\alpha^{\langle i \rangle}} = \varnothing$. If the height of the encoding tree is restricted to $K$, it is called a $K$-level encoding tree.*

**Definition 2.** *Structural Entropy. Based on the above definition, structural entropy can be used to quantify the dynamic complexity of such hierarchical trees, revealing their inherent topology insights. For the $K$-height encoding tree $\mathcal{T}$, the $K$-dimensional structural entropy of $\mathcal{G}$ is defined as:*

$$\mathcal{H}^K(\mathcal{G}) = \min_{\forall \mathcal{T}:height(\mathcal{T}) \leq K} \{\mathcal{H}^{\mathcal{T}}(\mathcal{G})\} \tag{8}$$

$$\mathcal{H}^{\mathcal{T}}(\mathcal{G}) = \sum_{\alpha \in \mathcal{T}, \alpha \neq \lambda} \mathcal{H}^{\mathcal{T}}(\mathcal{G}, \alpha) = -\sum_{\alpha \in \mathcal{T}, \alpha \neq \lambda} \frac{g_\alpha}{\mathrm{vol}(\mathcal{G})} \log_2 \frac{\mathrm{vol}(\alpha)}{\mathrm{vol}(\alpha^+)} \tag{9}$$

*where $g_\alpha$ represents the sum of the weights of cross-node edges that connect nodes within partition $\mathcal{C}_\alpha$ to nodes outside $\mathcal{C}_\alpha$. $\mathrm{vol}(\alpha)$ denotes the sum of the degrees of all nodes within $\mathcal{C}_\alpha$. $\mathcal{H}^{\mathcal{T}}(\mathcal{G}, \alpha)$ is the structural entropy of node $\alpha$, representing the $K$-dimensional structural entropy of $\alpha$ when the height of $\mathcal{T}$ is $K$. The core of this measurement lies in the observation that, in a highly connected graph, nodes frequently interact with their neighbors. By employing random walks, these interactions can be captured, and entropy can be introduced as a measure of topology uncertainty Li & Pan (2016). Specifically, the one-dimensional structural measurement of $\mathcal{G}$ can be quantified using the stationary distribution of its degrees $d$ and Shannon entropy. This concept can be generalized to $K$-dimensional measurements using the structural encoding tree in Definition 1.*

## B  ANALYSIS OF HIERARCHICAL COMMUNITY

In the encoding tree, the communities of leaf nodes are referred to as low-level communities, which are directly instantiated through the parent nodes of the leaf nodes. In contrast, communities at higher levels are referred to as higher-level communities. Low-level communities primarily capture local topological structures, often exhibiting homophily (i.e., connected nodes are more likely to share similar feature distributions or the same labels.) in TAGs, and represent local clusters or groups within the graph. These communities typically include strongly interrelated nodes, such as teams or social groups. Higher-level communities, on the other hand, are more abstract and loosely connected, formed by aggregating multiple low-level communities. They reveal high-level structural patterns or inter-community relationships, focus more on capturing global topological structures.

In Fig. 6, we present a toy example illustrating hierarchical semantics in a social network. It represents student social interactions, where nodes correspond to individual students, and edges denote relationships between them. Nodes of the same color represent students belonging to the same class (C-1, C-2, C-3), while red and blue nodes represent students sharing the same major (M-1), and

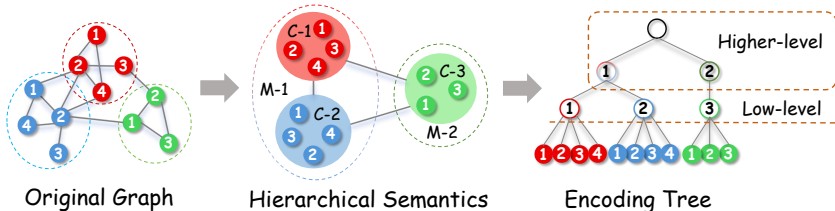

Figure 6: An illustrative example of hierarchical communities (semantics) in a simple social network.

green nodes represent students from a different major (M-2). The structural encoding tree reveals that low-level communities correspond to tightly connected classmates, while higher-level communities represent majors encompassing multiple classes. Although connections exist from different classes, they are less cohesive than those within the same class.

In LLaTA, we focus on low-level communities as they capture local relationships between nodes, which are critical for specific downstream tasks. Specifically, these communities reveal clearer and more concrete structural patterns, making them directly applicable to enhance GSL. In contrast, higher-level communities often represent latent high-level structural patterns, which is unclear and unintuitive. This makes them not well-suited for direct application in GSL. Furthermore, unlike naive neighborhood-based clusters, bottom-level communities capture local structural patterns while being shaped by global structural constraints.

While low-level communities are primarily focused on capturing local topological structure, they are not simple local neighborhoods. Under the global structural entropy minimization principleLi & Pan (2016), the construction of the encoding tree ensures that each module, including those at the bottom level, is formed in a way that minimizes global uncertainty and boundary complexity.

From the random walk perspective, the structural entropy $\mathcal{H}^{\mathcal{T}}(\mathcal{G})$ quantifies the average uncertainty of a walker's position in the network under a hierarchical modular encoding. As shown in Li & Pan (2016), the structural entropy with respect to a flat partition $\mathcal{P}$ can be decomposed as:

$$\mathcal{H}^{\mathcal{P}}(\mathcal{G}) = \sum_{\phi \in \mathcal{T}} \frac{\mathrm{vol}(\phi)}{\mathrm{vol}(\mathcal{G})} \cdot H\left(\left\{\frac{d_i^{(\phi)}}{\mathrm{vol}(\phi)}\right\}_{i=1}^{n_\phi}\right) - \sum_{\phi \in \mathcal{T}} \frac{g_\phi}{\mathrm{vol}(\mathcal{G})} \log_2 \frac{\mathrm{vol}(\phi)}{\mathrm{vol}(\mathcal{G})} \tag{10}$$

where $\mathrm{vol}(\phi) = \sum_{v \in \mathcal{C}_\phi} d_v$ is the volume of community $\mathcal{C}_\phi$, $d_i^{(\phi)}$ is the degree of node $i$ in $\mathcal{C}_\phi$ and $g_\phi$ is the sum of degrees crossing from $\mathcal{C}_\phi$ to other communities.

The first term is the **intra-community entropy**, describing the uncertainty of locating a node within its community given the walker is in $\mathcal{C}_\phi$:

$$H\left(\left\{\frac{d_i^{(\phi)}}{\mathrm{vol}(\phi)}\right\}\right) = -\sum_{i=1}^{n_\phi} \frac{d_i^{(\phi)}}{\mathrm{vol}(\phi)} \log_2 \frac{d_i^{(\phi)}}{\mathrm{vol}(\phi)} \tag{11}$$

The second term measures the **inter-community entropy**, i.e., the cost of describing which community the walker is in. This includes a penalty for community boundary size $g_\phi$, effectively discouraging communities with weak modularity or noisy boundaries.

Thus, during entropy minimization, a low-level community $\mathcal{C}_\phi$ is chosen not merely to cluster adjacent nodes, but to: - Minimize internal entropy by concentrating walk probability on structurally coherent nodes; - Minimize boundary cost $g_\phi$, ensuring the community is isolated under random walk dynamics; - And globally reduce the overall coding length of paths on $\mathcal{G}$. This gives rise to low-level communities that are both **locally tight** and **globally consistent**, satisfying structural objectives beyond local adjacency. Importantly, since $g_\phi$ and $\mathrm{vol}(\phi)$ are both computed relative to $\mathrm{vol}(\mathcal{G})$, each community even at the low-level is evaluated in a **global context**. Hence, low-level communities are globally optimized compression units of random walk behavior, fundamentally distinct from simple neighborhood clusters.

In summary, we leverage the low-level communities of the encoding tree as the basis for high-quality topological context construction, and perform leaf-level sampling within them to facilitate graph structure optimization with both local and global insight.

## C    EXISTING GSL PARADIGMS AND METHODS

### C.1    EXISTING GSL PARADIGMS

The existing graph structure learning models can be broadly classified into two paradigms: **Coupled** and **Decoupled**. Below, we will analyze these two paradigms and discuss which one should be adopted for new graph structure learning in the era of LLMs.

**Coupled Paradigm**. The graph learner and backbone are tightly coupled Chen et al. (2020); Fatemi et al. (2021); Wu et al. (2023) to learn improved structure by specific downstream tasks (e.g., node, link, and graph level). *Limitations*: *(1) Unstable Performance*: This task- and backbone-specific architecture limits the generalizability of the improved structure. Moreover, the strong dependency between the structure learning module and the downstream GNN makes performance vulnerable to low-quality graphs: if the original graph is poor, the resulting low-quality embeddings from the GNN directly degrade the effectiveness of the structure learning module. *(2) Inflexible Deployment*: Switching the downstream backbone or task requires retraining, which significantly limits its adaptability and scalability.

**Decoupled Paradigm**. The graph learner and downstream backbone are trained independently Liu et al. (2022b;a); Bi et al. (2024), avoiding co-training, thereby enabling better compatibility with LLMs for GSL. We observe that recent LLM-based GSL methods also tend to adopt this decoupled paradigm Guo et al. (2024); Zhang et al. (2024). *However, these methods exhibit a different form of coupling (i.e., fine-tuning based on the pre-defined instruction datasets), which leads to the following issues*: (1) They fail to directly and jointly incorporate topology and text, relying instead on instruction datasets. Moreover, they are highly sensitive to the quality of these datasets. (2) They rely on complex mechanisms and significant time to construct instruction datasets and fine-tuning. Therefore, from a model design perspective, we recommend adopting a comprehensive decoupled paradigm to integrate GSL and LLMs, minimizing reliance on fine-tuning and instead prioritizing more efficient and reliable inference with high-quality in-context prompts.

### C.2    COMPARATIVE ANALYSIS WITH SE-GSL AND LLM-BASED METHODS

In this section, we provide a detailed comparative analysis between our proposed LLaTA and existing approaches, including SE-GSL Zou et al. (2023), GraphEdit Guo et al. (2024), LLM4RGNN Zhang et al. (2024), and LangGSL Su et al. (2024). The goal is to highlight the fundamental differences in design and demonstrate the advantages of our method.

**SE-GSL.** SE-GSL follows a *coupled paradigm*, where a GNN is first used to obtain node embeddings, and the graph is then enhanced based on the similarity between embeddings. An encoding tree is subsequently constructed on the enhanced graph, followed by sampling to recover the structure. The updated graph is then fed back into the GNN for retraining, forming an iterative loop. In contrast, our LLaTA adopts a **decoupled paradigm**: we directly construct the encoding tree on the original graph and reformulate graph structure learning as an encoding tree optimization task. By leveraging the encoding tree to represent multi-level community structures, we enable the LLM to generate embeddings that integrate textual and structural information, which then guide the refinement of the tree and update the graph structure, all without retraining or coupling with a specific backbone.

**LLM-based Methods.** GraphEdit and LLM4RGNN both rely on fine-tuned LLMs to directly infer edge existence. GraphEdit improves connectivity through an edge predictor and LLM-guided node relevance evaluation, while LLM4RGNN emphasizes adversarial robustness by detecting malicious edges and predicting missing ones. However, both approaches suffer from high computational overhead since edge-level inference scales with the number of edges, which grows rapidly with graph size. Moreover, their edge-level predictions often ignore higher-order structural constraints, making the optimized graphs less consistent with the original community structure. LangGSL takes a different route by jointly optimizing node features and graph structures through an end-to-end framework where LLMs denoise attributes and GSL models refine connectivity. While effective, this

approach remains tied to specific GSL backbones and requires additional training, limiting scalability and generalization across diverse settings. In contrast, our LLaTA reformulates GSL as an **encoding tree optimization task**, where the tree provides a compact, hierarchical representation of community structures. This allows us to exploit LLMs for generating embeddings that integrate textual semantics and structural context at the *node/community level* rather than the *edge level*. As a result, our method is significantly more efficient, since the inference complexity depends on the number of nodes instead of the number of edges. Furthermore, by aligning edge updates with the optimized encoding tree, LLaTA produces graph structures that are more faithful to the original topology, thereby improving both robustness and effectiveness in downstream tasks. Unlike prior methods, LLaTA achieves these benefits in a **training-free and backbone-agnostic** manner, making it more practical and scalable.

**Summary of Advantages.** Overall, our LLaTA introduces several innovations that distinguish it from prior work. (1) It reformulates GSL as a **tree optimization task**, rather than relying on iterative retraining or direct edge prediction. (2) It is **training-free and decoupled**, requiring no fine-tuning of LLMs or dependency on specific GNN backbones. (3) It achieves **higher efficiency** by shifting complexity from the edge level to the node/community level, significantly reducing inference cost. (4) By refining the encoding tree, it ensures that structural updates **respect the original community divisions**, leading to improved robustness, interpretability, and generalization.

## D  EMPIRICAL STUDY

In this section, we demonstrate why our proposed tree-based GSL optimization pipeline outperforms other well-trained edge predictor-based methods through node classification experiments on the Citeseer dataset. Specifically, we compare the structure improvement quality of LLaTA against 7 prevalent traditional GSL methods and 3 recent LLM-based GSL methods using 3 metrics: Accuracy, Over-Smoothing, and Over-Squashing.

**Accuracy**. An intuitive method to evaluate the quality of the improved graph structure is to directly compare the downstream performance. In our implementation, we use a 2-layer GCN to perform node classification, with the resulting accuracy serving as a measure of improved structure.

**Over-smoothing and Over-squashing**. They are two common challenges in graph learning Nguyen et al. (2023); Zheng et al. (2024). Specifically, over-smoothing occurs when node features become indistinguishable as they converge to similar values, while over-squashing arises when nodes fail to capture information from distant neighbors, particularly in the presence of local bottlenecks in the graph. Both issues are strongly tied to the quality of the graph structure. To evaluate graph structure quality in light of these challenges, we employ the following analysis method: To begin with, we train a GCN on the improved graph to generate node embeddings. Then, we compute over-smoothing and over-squashing values by randomly sampling several node pairs: (1) Over-smoothing: We randomly sample $0.2 \times |\mathcal{V}|$ pairs of heterophilous nodes and compute their average cosine similarity based on the GCN-generated node embeddings. (2) Over-squashing: We randomly sample $0.2 \times |\mathcal{V}|$ pairs of distant homophilous nodes and compute their average cosine similarity using the same embeddings.

The experimental results are presented in Fig.1(d) in Sec.3 of the main text. In our reports, higher accuracy ($\uparrow$) reflects the improved performance of the downstream GNN on the enhanced graph. Lower over-smoothing ($\downarrow$) values indicate that the improved graph better distinguishes nodes of different classes, enabling the GNN to learn more discriminative node embeddings. Higher over-squashing ($\uparrow$) values suggest that the improved graph establishes more connections between distant homophilous nodes, enhancing the model's ability to capture information from long-range node pairs. According to the experimental results, our proposed decoupled and training-free LLaTA achieves the best performance across all three metrics, demonstrating the effectiveness of the tree-based GSL optimization pipeline compared to other well-trained edge predictor-based methods. Furthermore, we observe that LLM-based GSL methods often outperform traditional GSL approaches, highlighting the importance of developing effective GSL frameworks with text-processing capabilities in the era of LLMs for TAGs.

# E   ALGORITHM

## E.1   STRUCTURAL ENTROPY MINIMIZATION

To effectively capture graph topology, we introduce a structural entropy minimization algorithm to construct a structural encoding tree. This tree reorganizes the graph hierarchically, simulating its topology evolution. Based on this, we generate tree-based high-quality prompts for reliable inference without fine-tuning. This section presents the proposed algorithm and defines its three fundamental operations as follows:

**Definition 3.  Node Initializing.**  *Consider a node-wise TAG $\mathcal{G} = (\mathcal{V}, \mathcal{E})$ and a root node $\lambda$ in structural encoding tree $\mathcal{T}$. Let $v$ be a node in $\mathcal{G}$. Node Initializing $\mathrm{NI}_{\mathcal{T}}(v, \alpha)$ create node $\alpha$ for $v$ with $\mathcal{C}_{\alpha} = v$ and $\alpha^{+} = \lambda$.*

**Definition 4.  Node Combining.**  *Consider an encoding tree $\mathcal{T}$ for $\mathcal{G} = (\mathcal{V}, \mathcal{E})$, and let $\alpha$ and $\beta$ be two nodes in $\mathcal{T}$ that share the same parent $\gamma$. Node combining $\mathrm{NC}_{\mathcal{T}}(\alpha, \beta)$ can be represented as: $\gamma \leftarrow \phi^{+}; \phi \leftarrow \alpha^{+}; \phi \leftarrow \beta^{+}$. Here, $\phi$ replaces $\gamma$ as the new parent of $\alpha$ and $\beta$.*

**Definition 5.  Node Lifting.**  *Consider an encoding tree $\mathcal{T}$ for $\mathcal{G} = (\mathcal{V}, \mathcal{E})$, and let $\alpha$, $\beta$ and $\gamma$ be the nodes in $\mathcal{T}$, satisfying $\alpha^{+} = \beta$ and $\beta^{+} = \gamma$. Node Lifting $\mathrm{NL}_{\mathcal{T}}(\alpha, \beta)$ can be represented as: $\gamma \leftarrow \alpha^{+}$; IF $\mathcal{C}_{\beta} = \varnothing$, `delete`$(\beta)$. Here, $\alpha$ is lifted to the same height as its parent node $\beta$, and if $\beta$ has no children after the lifting, it is removed from $\mathcal{T}$.*

Based on this, the core of our employed greedy algorithm is as follows: (1) Initialization. The structural encoding tree is initialized with 1 height. Then, a root node is created and each original graph node is a leaf node. (2) Node Combination. Pairs of nodes in the tree are iteratively combined to minimize $\mathcal{H}^{\mathcal{T}}(\mathcal{G})$ at each step. This process continues until the root node has at most two children. (3) Node Lifting. Nodes are iteratively lifted to minimize $\mathcal{H}^{\mathcal{T}}(\mathcal{G})$ after each operation. This process is repeated until the tree height is reduced to $K$, resulting in a final encoding tree $\mathcal{T}^{K}$ with height $K$.

The pseudo-code of the high-dimensional structural entropy minimization algorithm is shown in Algorithm E.1.

---

**Algorithm 1:** Structural Entropy Minimization

1  **Input:** A graph $\mathcal{G}$, the height of encoding tree $K$.
2  **Output:** Encoding tree $\mathcal{T}^{K}$ with height $K$.
3  // Initialize encoding tree $\mathcal{T}$ with height 1
4  Create root node $\lambda$;
5  **for** $v_i \in \mathcal{V}$ **do**
6      $\lfloor$ $\mathrm{NI}_{\mathcal{T}}(v_i, \alpha_i)$ according to Definition. 3;
7  // Node combining
8  **while** $\lambda$ *has more than 2 children* **do**
9      Select $\alpha$ and $\beta$ in $\mathcal{T}$, conditioned on $\alpha^{+} = \beta^{+} = \lambda$ and
$$\arg\max_{\alpha, \beta} \left( \mathcal{H}^{\mathcal{T}}(\mathcal{G}) - \mathcal{H}^{\mathcal{T}}_{\mathrm{NC}_{\mathcal{T}}(\alpha, \beta)}(\mathcal{G}) \right);$$
10     $\lfloor$ $\mathrm{NC}_{\mathcal{T}}(\alpha, \beta)$ according to Definition. 4;
11 // Node lifting
12 **while** $height(\mathcal{T}) > K$ **do**
13     Select non-root nodes $\alpha$ and $\beta$ in $\mathcal{T}$, conditioned on $\alpha^{+} = \beta$ and
14     $$\arg\max_{\alpha, \beta} \left( \mathcal{H}^{\mathcal{T}}(\mathcal{G}) - \mathcal{H}^{\mathcal{T}}_{\mathrm{NL}_{\mathcal{T}}(\alpha, \beta)}(\mathcal{G}) \right);$$
15     $\lfloor$ $\mathrm{NL}_{\mathcal{T}}(\alpha, \beta)$ according to Definition. 5;
16 **Return** $\mathcal{T}^{K} \leftarrow \mathcal{T}$;

---

The structural entropy minimization algorithm reorganizes the graph into a hierarchical encoding tree, minimizing structural entropy to capture topological patterns. Through operations like Node Initializing, Node Combining, and Node Lifting, the algorithm refines the tree by combining nodes and adjusting their positions to reduce entropy. The final tree provides high-quality prompts for reliable inference, enhancing graph structure learning without fine-tuning, and improving efficiency for various tasks.

## E.2 TREE-BASED ADAPTIVE CLUSTERING

To optimize the structural encoding tree obtained by Appendix E.1 using text-driven node features, we introduce an adaptive clustering approach based on tree structures and the silhouette coefficient. This method reallocates leaf nodes according to text-driven insights from LLMs. By maximizing homophily within communities, we ensure that nodes with similar label classes are effectively grouped, aligning with structural patterns commonly observed in real-world applications. For graphs exhibiting heterophily, which are less common but still present, recent studies have demonstrated that uncovering their intrinsic higher-order homophilous patterns is an effective strategy Dai et al. (2022); Du et al. (2022); Song et al. (2023).

Specifically, we first initialize the leaf-community clusters using the original structural encoding tree. Based on this, we perform adaptive clustering by silhouette coefficient. The core intuition is to maximize homophily within low-level communities, ensuring that nodes within the same community predominantly belong to the same label class. In other words, (1) it considers the majority label classes within each low-level community, reallocating minority-class leaf nodes; and (2) it leverages the silhouette coefficient to adaptively increase clusters, thereby creating new low-level communities that enhance the tree's ability to reveal potential homophily patterns.

The following part provides a detailed explanation of this adaptive clustering approach, which enhances the tree's capability to uncover potential homophily patterns. The pseudo-code of the proposed tree-based adaptive clustering algorithm is presented in Algorithm E.2.

---

**Algorithm 2:** Tree-based Adaptive Clustering

1 **Input:** Encoding tree $\mathcal{T}$, soft labels $\mathbf{Y}^{cls}$ for each leaf node, hyperparameter $s$.
2 **Output:** Optimized encoding tree $\mathcal{T}^{\star}$.
3 **for** $\mathcal{C}^{\ell} \in \mathcal{T}$ **do**
4     Let $\mathbf{Y}^{cls}_{\mathcal{C}^{\ell}} = \{y_1^{cls}, y_2^{cls}, \ldots, y_m^{cls}\}$ be the soft labels of all nodes in $\mathcal{C}^{\ell}$;
5     Let $k^{\star} = 0$ and $sil_{max} = -1$;
6     **for** $2 < k < \frac{m}{2}$ **do**
7        Perform k-means($\mathbf{Y}^{cls}_{\mathcal{C}^{\ell}}$, $k$) and calculate average silhouette coefficient $sil_k$ of all clusters;
8        **if** $sil_k - sil_{max} < s$ **then**
9           break;
10        **else**
11           $sil_{max} = sil_k$; $k^{\star} = k$;

12     Perform k-means($\mathbf{Y}^{cls}_{\mathcal{C}^{\ell}}$, $k^{\star}$) and get a set of clusters $\{C_1, C_2, \ldots, C_{k^{\star}}\}$;
13     **for** $C_i \in \{C_1, C_2, \ldots, C_{k^{\star}}\}$ **do**
14        **if** $|C_i| > 1$ **then**
15           Construct the leaf nodes in $C_i$ as a new community and set the parent node of the new community to be the same as the parent node of $\mathcal{C}^{\ell}$;
16        **else**
17           Reallocate the leaf node in $C_i$ to other low-level communities based on $\mathbf{Y}^{cls}$;

18 **Return** $\mathcal{T}^{\star} \leftarrow \mathcal{T}$;

---

In Algorithm E.2, $\mathcal{C}^{\ell}$ is the low-level community, $m$ is the number of nodes in $\mathcal{C}^{\ell}$, $C_i$ denotes the $i$-th cluster in the k-means result, $k^{\star}$ is the best number of clusters, $sil_{max}$ is the current maximum average silhouette coefficient, and $s$ is a hyperparamter and is typically a small value.

Based on the adaptive clustering approach described above, we have optimized the structural encoding tree by reallocating leaf nodes using text-driven node features. In experiments, this method significantly enhanced the homophily within the graph structure. By maximizing homophily within communities, we effectively grouped nodes with similar label classes, improving the expressiveness of the graph structure and resulting in higher accuracy and generalization in graph learning tasks. Specifically, for graphs exhibiting heterophily, we successfully modeled the higher-order homophilic patterns, demonstrating the method's robustness in handling complex graph structures.

## E.3  LEAF-ORIENTED TWO-STEP SAMPLING

To efficiently reconstruct edge connections from the tree, we propose leaf-oriented two-step sampling. This approach balances running efficiency and practical performance by carefully selecting the subset of leaf nodes. Specifically, this method consists of two key phases: first, it identifies nodes that require optimization by analyzing their topological properties within the graph structure; second, it selects candidate nodes based on semantic similarity to enhance connectivity. By strategically adding or removing edges between these nodes, the method enables effective, training-free GSL, enhancing the representation quality without additional model fine-tuning. The pseudo-code of the proposed leaf-oriented two-step sampling method is detailed in Algorithm E.3.

---

**Algorithm 3:** Leaf-oriented Two-step Sampling

---

1 **Input:** Original Graph $\mathcal{G}$, Optimized encoding tree $\mathcal{T}^\star$, soft label $\mathbf{Y}^{cls}$ for each leaf node, hyperparameters $\theta, r$.

2 **Output:** Optimized graph $\mathcal{G}^\star$.

3 **for** $\mathcal{C}^\ell \in \mathcal{T}^\star$ **do**

4     **for** $i = 1$ *to* $m \times r$ **do**

5         // Step 1: Sample a leaf node $\alpha$ from $\mathcal{C}^\ell$

6         Let $set_1 = \{\alpha | \alpha \in \mathcal{C}^\ell\}$;

7         **for** $\alpha \in set_1$ **do**

8             Compute $\mathcal{H}^{\mathcal{T}^\star}(\mathcal{G}, \alpha)$ according to Eq. (9);

9             Compute $P_{topo}(\alpha)$ according to Eq. (7);

10         Sample a leaf node $\alpha$ based on $P_{topo}(\alpha)$ from $set_1$;

11         // Step 2: Sample an edge for $\alpha$

12         Let $set_2 = \{\beta | \beta \in \mathcal{C}^\ell \text{ and } \beta \neq \alpha\}$;

13         **if** $|set_2| < \theta$ **then**

14             // Expand the candidate set

15             Let $\phi$ be the grandparent of $\alpha$;

16             **for** $\gamma \in \phi$ *and* $\gamma \neq \alpha^+$ **do**

17                 Compute $y_\gamma^{cls} = \frac{1}{m} \sum_{\alpha \in \gamma} y_\alpha^{cls}$;

18                 Compute $\text{sim}(y_\gamma^{cls}, y_\alpha^{cls})$ according to Eq. (3);

19             **for** $|set_2| < \theta$ **do**

20                 Choose low-level community with highest sim and add its children to $set_2$;

21         **for** $\beta \in set_2$ **do**

22             Compute $\text{sim}(y_\beta^{cls}, y_\alpha^{cls})$ according to Eq. (3);

23             Compute $P_{sema}^\alpha(\beta)$ according to Eq. (7);

24         **if** $|set_2| > \theta$ **then**

25             Keep the top $\theta$ nodes in $set_2$ based on the soft label similarity with $\alpha$.

26         For edge addition, sample a leaf node $\beta$ from $set_2$ based on $P_{sema}^\alpha(\beta)$ ranked in descending order;

27         For edge removal, sample a leaf node $\beta$ from $set_2$ based on $P_{sema}^\alpha(\beta)$ ranked in ascending order;

28         Add edge to or remove edge from graph $\mathcal{G}$;

29 **Return** $\mathcal{G}^\star \leftarrow \mathcal{G}$;

---

The leaf-oriented two-step sampling method optimizes edge connections by first identifying nodes that require optimization based on their topological properties, ensuring that critical parts of the graph are prioritized for enhancement. In the second phase, candidate nodes are selected using semantic similarity to improve connectivity, focusing on nodes that are contextually relevant. This approach strikes a balance between computational efficiency and practical performance, enabling effective graph structure learning without the need for training. By incorporating both topological and semantic insights, the method refines the graph's structure, improving its representation quality while ensuring robustness, adaptability, and minimal computational overhead for various graph tasks.

# F   THEOREM PROOF

In this section, we provide proofs for all theorems stated in the main text.

## F.1   TOPOLOGICAL INFORMATION CAPTURING CAPABILITY OF THE ENCODING TREE

**Theorem 1** (Topological Information Capturing Properties of Encoding Tree). *Given an encoding tree $\mathcal{T}$ and a non-leaf node $\phi \in \mathcal{T}$, the error of topological information $\varepsilon^h(\phi)$ in community $\mathcal{C}_\phi$ is upper bounded by: $\frac{g_\phi}{2m} \log_2 \frac{\text{vol}(\phi^+)}{g_\phi}$, and $\varepsilon^h(\phi)$ gradually decreases as the community level descends.*

*Proof.* We partition the proof into two components: (i) derivation of the upper bound for topological information error, and (ii) demonstration of error variation across community hierarchy levels.

**Part 1: Upper Bound on Topological Information Error**

Suppose a network $\mathcal{G} = (\mathcal{V}, \mathcal{E})$ is given, along with a encoding tree $\mathcal{T}$ of height $K$. Let $\phi$ be an non-leaf node (community) in $\mathcal{T}$ located at level $k$, and denote the corresponding community as $\mathcal{V}_\phi \subseteq V$. Let $\mathcal{G}_\phi = (\mathcal{V}_\phi, \mathcal{E}_\phi)$ be the actual subgraph induced by the community $\mathcal{C}_\phi$. Then, the *topological information error* $\varepsilon^h(\phi)$ is defined as the absolute difference between the true structural entropy of the subgraph $\mathcal{G}_\phi$ and the approximate structural entropy derived from the encoding tree representation:

$$\varepsilon^h(\phi) = \left| \mathcal{H}(\mathcal{G}_\phi) - \mathcal{H}^{\mathcal{T}}(\mathcal{G}, \phi) \right|, \tag{12}$$

where $\mathcal{H}(\mathcal{G}_\phi)$ denotes the optimal value of structural entropy for the true subgraph $\mathcal{G}_\phi$, $\mathcal{H}^{\mathcal{T}}(\mathcal{G}, \phi)$ denotes the local structural entropy corresponding to community $\phi$ as defined by the encoding tree $\mathcal{T}$.

First, since the true structural entropy $\mathcal{H}(\mathcal{G}_\phi)$ is defined based on the optimal internal sub-community partitioning, it must be less than or equal to the entropy under a completely random structure. In the worst case, if the internal structure of community $\phi$ is entirely random, the upper bound of its true structural entropy is:

$$\mathcal{H}(\mathcal{G}_\phi) \leq -\frac{g_\phi}{2m} \log_2 \frac{g_\phi}{\text{vol}(\phi^+)} \tag{13}$$

The intuition here is that when the internal structure of community $\phi$ is entirely random, the uncertainty of a random walk entering community $\phi$ reaches its maximum, resulting in the highest possible entropy.

On the other hand, the structural entropy defined by the encoding tree $\mathcal{T}$ is given by:

$$\mathcal{H}_{\mathcal{T}}(\mathcal{G}; \phi) = -\frac{g_\phi}{2m} \log_2 \frac{\text{vol}(\phi)}{\text{vol}(\phi^+)} \tag{14}$$

Now, we consider the worst-case scenario, that is, the difference between the maximally random structure (with highest entropy) and the simplified approximation provided by the tree (with smaller entropy). Substituting the expressions in Eq. 12 according to Eq. 13 and Eq. 14, we obtain:

$$\varepsilon^h(\phi) = |\mathcal{H}(\mathcal{G}_\phi) - H_{\mathcal{T}}(\mathcal{G}; \phi)| \leq \left| -\frac{g_\phi}{2m} \log_2 \frac{g_\phi}{\text{vol}(\phi^+)} + \frac{g_\phi}{2m} \log_2 \frac{\text{vol}(\phi)}{\text{vol}(\phi^+)} \right| \tag{15}$$

Simplifying the expression, we get:

$$\varepsilon^h(\phi) \leq \frac{g_\phi}{2m} \left| \log_2 \frac{\text{vol}(\phi)}{g_\phi} \right| \tag{16}$$

Since for any community $\phi$, it always holds that $\text{vol}(\phi) \geq g_\phi$ (as the volume of a community must be greater than or equal to the number of its external edges), we can further loosen the bound by replacing $\text{vol}(\phi)$ with the larger and more general parent volume $\text{vol}(\phi^+) \geq \text{vol}(\phi)$, yielding a broader upper bound:

$$\varepsilon^h(\phi) \leq \frac{g_\phi}{2m} \log_2 \frac{\text{vol}(\phi^+)}{g_\phi} \tag{17}$$

This forms the desired expression for the upper bound of the topological information error in the Theorem. 1.

**Part 2: Error Decay Across Hierarchical Community Levels**

In this part, we prove that the topological information error becomes progressively smaller as we descend into lower layers (finer communities) of the partitioning tree.

**Lemma 1** (Locality and Additivity of Structural EntropyLi & Pan (2016)). *Let $\mathcal{G} = (\mathcal{V}, \mathcal{E})$ be a network and let $\mathcal{T}$ be a hierarchical encoding tree of $\mathcal{G}$. For any non-leaf node $\phi \in \mathcal{T}$ with children $\beta_1, \ldots, \beta_c$, the structural entropy of the subgraph $\mathcal{G}_\phi$ induced by community $\mathcal{C}_\phi$ satisfies:*

$$\mathcal{H}(\mathcal{G}_\phi) = \sum_{i=1}^{c} \mathcal{H}(\mathcal{G}_{\beta_i}) + \mathcal{H}^{\mathrm{cross}}(\phi), \quad \mathcal{H}^{\mathrm{cross}}(\phi) = -\sum_{i=1}^{c} \frac{g_{\beta_i}}{2m} \log_2 \frac{\mathrm{vol}(\beta_i)}{\mathrm{vol}(\phi)} \tag{18}$$

*where $\mathcal{H}^{\mathrm{cross}}(\phi)$ is the cross-community contribution that accounts for the uncertainty incurred when identifying sub-communities within $\mathcal{C}_\phi$.*

Let $\phi$ be a community at height $k$ in the tree $\mathcal{T}$, and let its children be $\beta_1, \ldots, \beta_c$. In Eq. 12, we define the topological information error as: $\varepsilon^h(\phi) = |\mathcal{H}(\mathcal{G}_\phi) - H_\mathcal{T}(\mathcal{G}, \phi)|$.

Expand the first term of the error using Lemma 1:

$$\varepsilon^h(\phi) = \left| \sum_{i=1}^{c} \mathcal{H}(\mathcal{G}_{\beta_i}) + \mathcal{H}^{\mathrm{cross}}(\phi) - H^\mathcal{T}(\mathcal{G}, \phi) \right|. \tag{19}$$

Applying the **triangle inequality**, we obtain an upper bound:

$$\varepsilon^h(\phi) \leq \left| \sum_{i=1}^{c} \left( \mathcal{H}(G_{\beta_i}) - H^\mathcal{T}(G; \beta_i) \right) \right| + \mathcal{H}^{\mathrm{cross}}(\phi). \tag{20}$$

To rigorously demonstrate that the error decreases as we move to lower levels (smaller $k$), we analyze the two terms contributing to the error:

1. **Local structure approximation error** $| \sum_{i=1}^{c} \left( \mathcal{H}(G_{\beta_i}) - H^\mathcal{T}(G; \beta_i) \right) |$**:**

   For each child community $\beta_i$, the local approximation error satisfies the following upper bound (as established in part 1):

$$\left| \mathcal{H}(G_{\beta_i}) - H^\mathcal{T}(G; \beta_i) \right| \leq \frac{g_{\beta_i}}{2m} \left| \log_2 \frac{\mathrm{vol}(\beta_i)}{g_{\beta_i}} \right|. \tag{21}$$

   As we move to lower-level communities (i.e., smaller $k$), both $\mathrm{vol}(\beta_i)$ and $g_{\beta_i}$ decrease due to finer partitions and reduced external connectivity. Thus, each term on the right-hand side of Eq. 21 becomes smaller, and so does their sum.

2. **Cross-community entropy term** $\mathcal{H}^{\mathrm{cross}}(\phi)$**:**

   Recall that the cross-community entropy term in Eq. 18, as we go deeper in the tree (i.e., lower communities), both the volumes and $g_{\beta_i}$ shrink, making this term smaller. Hence, $\mathcal{H}^{\mathrm{cross}}(\phi)$ also decreases as the community level descends.

Combining the above, since both $| \sum_{i=1}^{c} \left( \mathcal{H}(G_{\beta_i}) - H^\mathcal{T}(G; \beta_i) \right) |$ and $\mathcal{H}^{\mathrm{cross}}(\phi)$ tend to decrease as we move to lower-level communities in the encoding tree (i.e., toward finer communities), the overall error $\varepsilon^h(\phi)$ also tends to decrease with depth. This supports the claim that topological information error decays along the hierarchical levels. $\square$

## F.2 GLOBAL INFORMATION CONSTRAINTS OF THE LOW-LEVEL COMMUNITIES

**Theorem 2** (Implicit Global Constraints in Low-Level Communities). *In a structural encoding tree $\mathcal{T}$, each low-level community $\mathcal{C}^l$ captures localized topology but implicitly retains global structure constraints due to the hierarchical, random-walk-based formulation of structural entropy.*

*Proof.* We partition the proof into two components: (i) low-level communities capture localized topology, and (ii) low-level communities implicitly retains global structure constraints.

**Part 1: Low-level Communities Capture Localized Topology**

From the definition perspective, the structural entropy of a community $\mathcal{C}_\phi \in \mathcal{T}$ is defined as:

$$\mathcal{H}^{\mathcal{T}}(G, \phi) = -\frac{g_\phi}{2m} \log_2 \left( \frac{\text{vol}(\phi)}{\text{vol}(\phi^+)} \right), \tag{22}$$

where $g_\phi$ is the sum of the cross-node edges that connect nodes within $\phi$ to nodes outside $\phi$, $\text{vol}(\phi) = \sum_{v \in \phi} \deg(v)$ is the volume of $\phi$, and $\phi^+$ is the parent of $\phi$ in the tree. This expression depends only on the volume and cut-size between $\phi$ and $\phi^+$, rather than direct interaction with distant or global components of the graph.

Moreover, since the low-level community $\mathcal{C}_\phi^\ell$ typically satisfy:

$$\text{vol}(\phi) \ll \text{vol}(\phi^+), \quad \text{and} \quad g_\phi \ll g_\lambda = 2m, \tag{23}$$

the entropy term of the low-level community $\mathcal{C}_\phi^\ell$ becomes more localized Li & Pan (2016).

From the algorithmic perspective, the construction of $\mathcal{T}$ through structural entropy minimization encourages the formation of tightly connected, sparsely separated communities. As a result, communities near the leaves tend to consist of closely connected nodes with minimal external edges.

Therefore, low-level communities, both by definition and by optimization, are structurally biased toward encoding localized topological patterns.

**Part 2: Low-level Communities Implicitly Retains Global Structure Constraints**

We consider a graph $\mathcal{G} = (\mathcal{V}, \mathcal{E})$ and its stationary random walk process, where transition probabilities are defined by:

$$P_{uv} = \frac{\mathbf{A}(u, v)}{\deg(u)}, \quad \text{and} \quad \pi_u = \frac{\deg(u)}{2m}, \tag{24}$$

where $\mathbf{A}$ is adjacency matrix of $\mathcal{G}$ and $\deg$ is the degree of the node.

Let the encoding tree $\mathcal{T}$ be constructed by minimizing the structural entropy, which reflects the minimum cost of encoding a random walk trajectory under a hierarchical community structure.

The structural entropy is defined as:

$$\mathcal{H}^{\mathcal{T}}(\mathcal{G}) = \sum_{\phi \in \mathcal{T}, \phi \neq \lambda} \mathcal{H}^{\mathcal{T}}(\mathcal{G}, \phi) = \sum_{\phi \in \mathcal{T}, \phi \neq \lambda} -\frac{g_\phi}{2m} \log_2 \left( \frac{\text{vol}(\phi)}{\text{vol}(\phi^+)} \right). \tag{25}$$

Now let $\phi^{(k)} = \lambda, \phi^{(k-1)}, \ldots, \phi^{(1)} = \phi$ be the path in the tree from the root to a low-level community $\mathcal{C}_\phi^\ell$. Then the random walk conditional entry probability into $\phi$ given hierarchy is:

$$P(\text{enter } \phi) = \prod_{i=1}^{k-1} \frac{\text{vol}(\phi^{(i)})}{\text{vol}(\phi^{(i+1)})}. \tag{26}$$

This encodes the fact that the deeper a community lies, the more global decisions influence its encoding, each ratio $\frac{\text{vol}(\phi^{(i)})}{\text{vol}(\phi^{(i+1)})}$ reflects how the random walk transitions between communities at each level.

Thus, the total information needed to encode the walk into $\phi$ can be defined as the *path entropy*:

$$\mathcal{H}_{\text{path}}^{\mathcal{T}}(\phi) = -\frac{g_\phi}{2m} \log_2 \left( \frac{\text{vol}(\phi)}{\text{vol}(\phi^{(2)})} \right) + \cdots + -\frac{g_{\phi^{(1)}}}{2m} \log_2 \left( \frac{\text{vol}(\phi^{(k-1)})}{\text{vol}(\lambda)} \right) \tag{27}$$

This quantity corresponds to a subset of the full structural entropy $\mathcal{H}_{\mathcal{T}}^{\text{path}}(\alpha) \subseteq \mathcal{H}_{\mathcal{T}}(\mathcal{G})$, indicating that it reflects only the contribution of a specific path in the tree, while the full entropy aggregates over all nodes in $\mathcal{T}$.

From the perspective of path entropy, although only $\mathcal{H}^{\mathcal{T}}(\mathcal{G}, \phi)$ contributes directly to the total structural entropy in Eq. 25, the encoding of a random walk into $\phi$ must traverse the full ancestral path $\lambda \to \phi^{(k-1)} \to \cdots \to \phi$. Thus, the effective information cost of encoding a transition into $\phi$ is represented by $\mathcal{H}_{\text{path}}^{\mathcal{T}}(\phi)$, which includes contributions from all higher-level communities.

Moreover, since $\mathcal{T}$ is obtained by minimizing the total structural entropy $\mathcal{H}^{\mathcal{T}}(\mathcal{G})$, the $\mathcal{H}_{\text{path}}^{\mathcal{T}}(\phi)$ is also simultaneously subject to the minimization constraint, which means the division and refinement of communities—including low-level ones—are jointly optimized under global constraints. In particular, each local partition is chosen not independently, but in the context of its position within the tree, as reflected in equation Eq. 27.

Therefore, Even though the structure of $\phi$ is small and local, its encoding depends on the full sequence of volume and boundary terms from all ancestors $\phi^{(i)}$, which are determined by global topology. Hence, low-level communities capture local topological patterns while implicitly preserving global structural constraints. $\qquad\square$

### F.3   THE ERROR BOUND BETWEEN SOFT LABELS AND TRUE LABELS

**Theorem 3** (Error Bound Between Soft labels and True Labels). *Given two leaf nodes $\alpha$ and $\beta$ in $\mathcal{T}$, the error between soft label similarity and true label similarity is bounded by:*

$\varepsilon^y(\alpha\beta) = |\sim(y_\alpha^{cls}, y_\beta^{cls}) - \sim(y_\alpha, y_\beta)| \leq \delta \cdot (1 - \epsilon)$, *where $\delta$ is a constant that depends on the LLM's in-context learning ability, $y_\alpha$ is the true label of $\alpha$ and $\epsilon$ is the text similarity threshold.*

*Proof.* The proof leverages three key lemmas derived from the Transformer architecture in Vaswani et al. (2017).

**Lemma 2** (Attention-Induced Local Homophily). *Let $t_\alpha^\star$ be the aggregated text of node $\alpha$ using threshold $\epsilon$ (Eq. 2). The raw soft label error $\Delta_\alpha^{raw} = |y_\alpha^{cls} - y_\alpha|$ satisfies:*

$$\Delta_\alpha^{raw} \leq \delta_1 \cdot (1 - \epsilon), \tag{28}$$

*where $\delta_1$ depends on the attention weight distribution, and $\Delta_\alpha^{raw}$ is the raw prediction error before residual connections and layer normalization, directly generated by the output of the attention mechanism.*

*Proof.* From the scaled dot-product attention mechanism:

$$\text{Attention}(Q, K, V) = \text{softmax}\left( \frac{QK^T}{\sqrt{d_k}} \right) V, \tag{29}$$

the threshold $\epsilon$ filters neighbors with $w_{\alpha\beta} \geq \epsilon$ (Eq. 2 and Eq. 3). The multi-head attention reinforces local semantic consistency across subspaces, bounding the error proportionally to $(1 - \epsilon)$ Vaswani et al. (2017). $\qquad\square$

**Lemma 3** (Stability via Residual Connections). *The residual connections and layer normalization compress the error:*

$$\Delta_\alpha \leq \delta_2 \cdot \Delta_\alpha^{raw}, \tag{30}$$

*where $\delta_2 < 1$ is a normalization gain factor.*

*Proof.* The residual structure $\text{LayerNorm}(x + \text{Sublayer}(x))$ suppresses feature drift, with $d_{\text{model}}$ as the hidden dimension Vaswani et al. (2017):

$$\|\Delta_\alpha\| \leq \frac{|\Delta_\alpha^{\text{raw}}|}{\sqrt{d_{\text{model}}}} \leq \delta_2 \cdot |\Delta_\alpha^{\text{raw}}|, \tag{31}$$

where $d_{\text{model}}$ is the model dimension. The normalization operation stabilizes training by scaling the gradient direction, reducing the magnitude of error propagation. $\square$

**Lemma 4** (Representation Capacity). *The hidden dimension $d_{model}$ governs semantic precision:*

$$\delta_1 \propto d_{model}^{-0.5}. \tag{32}$$

*Proof.* The learning rate design $lrate = d_{\text{model}}^{-0.5} \cdot \min(\cdot)$ indicates that increasing the model dimensionality necessitates a reduction in learning rate to preserve training stability. A larger $d_{\text{model}}$ enhances the model's feature representation capacity, while the error decay rate per unit dimension scales proportionally with $d_{\text{model}}^{-0.5}$ Vaswani et al. (2017). $\square$

Recall the definition of cosine similarity, the similarity between two vectors is defined as:

$$\text{sim}(x_i, x_j) = \frac{x_i \cdot x_j}{\|x_i\|\|x_j\|} \tag{33}$$

Substituting this into our error analysis yields:

$$\varepsilon^y(\alpha\beta) = \left|\text{sim}(y_\alpha^{cls}, y_\beta^{cls}) - \text{sim}(y_\alpha, y_\beta)\right| = \left|\frac{y_\alpha^{cls} \cdot y_\beta^{cls}}{\|y_\alpha^{cls}\|\|y_\beta^{cls}\|} - y_\alpha \cdot y_\beta\right|. \tag{34}$$

Since the ground truth labels are one-hot encoded, satisfying $\|y_\alpha\| = \|y_\beta\| = 1$, the similarity can be simplified as: $y_\alpha \cdot y_\beta$.

Through **error decomposition** and application of the **triangle inequality**, we obtain:

$$\varepsilon^y(\alpha\beta) \leq \underbrace{\left|\frac{y_\alpha^{cls} \cdot y_\beta^{cls}}{\|y_\alpha^{cls}\|\|y_\beta^{cls}\|} - y_\alpha^{cls} \cdot y_\beta\right|}_{\text{Term 1}} + \underbrace{\left|y_\alpha^{cls} \cdot y_\beta - y_\alpha \cdot y_\beta\right|}_{\text{Term 2}}. \tag{35}$$

**Bounding Term 2:** Since $y_\beta$ is one-hot (non-zero at index $j$):

$$\text{Term 2} = \left|(y_\alpha^{cls} - y_\alpha) \cdot y_\beta\right| \leq \Delta_\alpha \cdot |y_\beta| \leq \delta_1\delta_2(1-\epsilon). \quad \text{(By Lemma 2\& 3.)} \tag{36}$$

**Lemma 5** (Normalization Perturbation). *For any vectors $u, v \in \mathbb{R}^d$ where $v$ is normalized ($\|v\| = 1$) and $\|u - v\| \leq \eta$, the following inequality holds for their normalized difference:*

$$\left\|\frac{u}{\|u\|} - v\right\| \leq \frac{2\eta}{\|u\|} \leq 2\eta$$

*where the second inequality follows from $\|u\| \geq 1 - \eta$ (by the reverse triangle inequality).*

**Bounding Term 1:** Using the vector normalization Lemma 5 with $u = y_\beta^{cls}$ and $v = y_\beta$. if $\|y_\beta^{cls} - y_\beta\| \leq \delta_1\delta_2(1-\epsilon) = \eta$, then:

$$\text{Term 1} \leq \|y_\alpha^{cls}\| \cdot \left\|\frac{y_\beta^{cls}}{\|y_\beta^{cls}\|} - y_\beta\right\| \leq 2\delta_1\delta_2(1-\epsilon). \tag{37}$$

**Final Bound:** Combining both terms and incorporating Lemma 4:

$$\varepsilon^y(\alpha\beta) \leq 3\delta_1\delta_2(1-\epsilon) \leq \delta \cdot (1-\epsilon), \tag{38}$$

where $\delta = 3\delta_1\delta_2 \cdot d_{\text{model}}^{-0.5}$ depends on the in-context learning ability of the LLMs.

In summary, we have proven that the similarity of soft labels $y^{cls}$ generated through our **Community of Thought** mechanism can approximate the similarity of ground-truth labels $y$ with controllable bias $\varepsilon^y$, where this bias decreases as:

- Local homophily threshold $\epsilon$ increases (i.e., tighter neighbor selection),

- and LLM in-context learning capacity $\delta$ improves (e.g., via larger model size or better training).

$\square$

### F.4 STRUCTURAL ENTROPY AND NODE TOPOLOGICAL UNCERTAINTY

**Theorem 4** (High-Entropy Nodes Require Supervision). *In a structural encoding tree $\mathcal{T}$ constructed via entropy minimization, nodes with higher structural entropy $\mathcal{H}^{\mathcal{T}}(\mathcal{G}, \alpha)$ indicate: higher topological uncertainty within their local structural context.*

*Proof.* We begin by recalling the definition of structural entropy for a node $\phi \in \mathcal{T}$, derived from the minimum encoding length of a random walk process:

$$\mathcal{H}^{\mathcal{T}}(\mathcal{G}, \phi) = -\frac{g_\phi}{2m} \log_2 \left( \frac{\text{vol}(\phi)}{\text{vol}(\phi^+)} \right), \tag{39}$$

where $g_\phi$ is the sum of the cross-node edges that connect nodes within $\phi$ to nodes outside $\phi$, $\text{vol}(\phi) = \sum_{v \in \phi} \deg(v)$ is the volume of $\phi$, and $\phi^+$ is the parent of $\phi$ in the tree.

Structural entropy encodes the uncertainty of random walk transitions. In the structural entropy formulation, the graph is viewed as a Markov chain, and each term $H^{\mathcal{T}}(\mathcal{G}, \phi)$ represents the expected number of bits needed to encode a random walker's transition from $\phi^+$ into $\phi$ along a hierarchical path in the encoding tree.

Let us consider the conditional transition probability from $\phi^+$ to $\phi$, denoted as:

$$P(\phi \mid \phi^+) = \frac{\text{vol}(\phi)}{\text{vol}(\phi^+)}. \tag{40}$$

The negative log of this transition probability reflects the information content (surprisal) of the walk entering $\phi$. When $\text{vol}(\phi)$ is close to $\text{vol}(\phi^+)$, i.e., $P(\phi \mid \phi^+) \to 1$, the encoding cost (entropy) becomes small. But when this ratio becomes ambiguous (e.g., many similar-volume subregions), entropy increases.

Moreover, the term $g_\phi$ acts as a weight reflecting how often the walker exits or enters $\phi$. A larger $g_\phi$ means more random walks intersect the boundary of $\phi$, suggesting weaker internal cohesion and a blurrier community boundary.

From the above, the entropy term $\mathcal{H}^{\mathcal{T}}(\mathcal{G}, \phi)$ grows when:

- $\frac{\text{vol}(\phi)}{\text{vol}(\phi^+)} \approx 0.5$, i.e., the walker has nearly equal chance to enter multiple subcommunities;

- $g_\phi$ is large, i.e., there are many transitions across community boundaries.

In such scenarios, the community $\phi$ has:

- a **high encoding cost** (random walk description length is large),

- and a **weak structural separability**, because the walker often crosses between $\alpha$ and other regions.

Thus, a node $v \in \mathcal{C}_\phi$ inherits this high entropy $\mathcal{H}_{\mathcal{T}}(\mathcal{G}, \phi)$, reflecting the fact that its structural position within the graph is more ambiguous. It is harder to determine whether it coherently belongs to a community based solely on graph structure, and hence it exhibits higher topological uncertainty. $\square$

Table 6: Details of experimental datasets.

| Dataset | # Nodes | # Edges | # Classes | # Train/Val/Test | # Homophily | Text Information | Domains |
|---|---|---|---|---|---|---|---|
| Cora | 2,708 | 10,556 | 7 | 60/20/20 | 0.81 | Title and Abstract of Paper | Citation |
| Citeseer | 3,186 | 8,450 | 6 | 60/20/20 | 0.74 | Title and Abstract of Paper | Citation |
| Pubmed | 19,717 | 88,648 | 3 | 60/20/20 | 0.80 | Title and Abstract of Paper | Citation |
| ArXiv | 169343 | 2315598 | 40 | 60/20/20 | 0.66 | Title and Abstract of Paper | Citation |
| WikiCS | 11,701 | 431,726 | 10 | 5/22.5/50 | 0.65 | Title and Abstract of Article | Knowledge |
| Instagram | 11,339 | 144,010 | 2 | 10/10/90 | 0.59 | Personal Profile of User | Social |
| Reddit | 33,434 | 269,442 | 2 | 10/10/90 | 0.59 | Last 3 Posts of User | Social |
| Ratings | 24,492 | 186,100 | 5 | 25/25/50 | 0.38 | Name of Product | E-commerce |
| Child | 23,327 | 240,604 | 12 | 60/20/20 | 0.42 | Name and Description of Book | E-commerce |
| History | 41,551 | 503,180 | 12 | 60/20/20 | 0.64 | Name and Description of Book | E-commerce |
| Photo | 48,362 | 873,793 | 12 | 60/20/20 | 0.74 | User Review of Product | E-commerce |

# G  DATASETS

We evaluate LLaTA and baselines on 11 widely adopted TAG datasets across multiple domains, including 4 citation networks Chen et al. (2024), 1 knowledge network Chen et al. (2024), 2 social networks Huang et al. (2024), and 4 e-commerce networks Yan et al. (2023). Among these, the Ratings and Child exhibit heterophily, while the remaining datasets follow homophily. The details of these TAG datasets are shown in Table 6. The description of the datasets for each domain is as follows:

- **Citation Networks**  Chen et al. (2024). Cora, Citeseer, Pubmed and ArXiv are benchmark datasets of citation networks. Nodes represent paper, and edges represent citation relationships. The features are obtained through pre-trained language model, and labels denote their academic fields.

- **Knowledge Networks**  Chen et al. (2024); Mernyei & Cangea (2020). WikiCS is a benchmark dataset of knowledge networks. Nodes represent articles in the field of computer science, and edges represent hyperlinks between these articles. The features are obtained through pre-trained language model, and labels denote different branches of computer science.

- **Social Networks**  Huang et al. (2024). Instagram and Reddit are benchmark datasets of social netowrks. Nodes represent users, and edges represent social relationhships. The features are obtained through pre-trained language model, and labels correspond to different types of users.

- **E-commerce Networks**  Yan et al. (2023). Ratings, Child, History and Photo are benchmark datasets of e-commerce networks. Nodes represent products, and edges represent relationships such as co-purchase or co-view. The features are obtained through pre-trained language model, and labels correspond to product categories or user ratings.

# H  BASELINES

In this section, we provide a brief description for each baseline used in the experiments. It includes 5 coupled GSL methods Chen et al. (2020); Fatemi et al. (2021); Zhao et al. (2021); Wu et al. (2023); Zou et al. (2023), 6 decoupled GSL methods Jin et al. (2020); Liu et al. (2022b); Li et al. (2022); Liu et al. (2022a); Nguyen et al. (2023); Bi et al. (2024), and 3 LLM-based GSL methodsGuo et al. (2024); Zhang et al. (2024). Based on this, we apply prevalent GNN (GCN Kipf & Welling (2016), GAT Veličković et al. (2017), GraphSAGE Hamilton et al. (2017)) and mainstream LLM-GNN (GLEM Zhao et al. (2022), ENGINE Zhu et al. (2024)) as the downstream backbone.

All baselines are briefly described as follows:

- **GCN** Kipf & Welling (2016). GCN is among the most widely adopted GNN architectures, as it introduces a first-order approximation of a localized spectral filter for graph-structured data.

- **GAT** Veličković et al. (2017). GAT incorporates a self-attention mechanism to compute importance scores for different neighboring nodes, enabling more effective aggregation of neighborhood information.

- **GraphSAGE** Hamilton et al. (2017). GraphSAGE [14] is an inductive framework that generates node embeddings by leveraging node features and using a sampling-based approach to aggregate information from the local neighborhood, enabling scalable representation learning.

- **GLEM** Zhao et al. (2022). GLEM proposes a scalable framework that integrates LLMs with GNNs through a variational EM algorithm. By alternating between optimizing LMs for textual semantics and GNNs for structural information, GLEM enables mutual distillation without costly end-to-end training, achieving promising performance on large-scale text-attributed graphs.

- **ENGINE** Zhu et al. (2024). ENGINE introduces an LLM-enhanced graph learning framework that combines LLMs with GNNs through a mutual integration mechanism. It leverages LLMs to enrich node representations with semantic knowledge while using GNNs to capture structural dependencies, and aligns the two via iterative refinement. This design improves both effectiveness and robustness on text-attributed graphs, showing promising performance across multiple benchmarks.

- **IDGL** Chen et al. (2020). IDGL is a framework that learns a refined graph structure by leveraging cosine similarity of features and top-k threshold refinement, combining it with the original graph for joint optimization.

- **SLAPS** Fatemi et al. (2021). SLAPS is a graph learning framework that uses MLP-based embeddings and kNN to build the graph. The adjacency matrix is symmetrized and normalized, and a self-supervised denoising autoencoder is used to update both the graph and model parameters.

- **GAUGO** Zhao et al. (2021). GAUGO uses a graph auto-encoder to learn a new graph structure. It predicts edges based on node features, refines the edge probabilities with Gumbel sampling, and fuses the generated graph with the original one before training. Both the generated graph and model parameters are updated during optimization.

- **HESGSL** Wu et al. (2023). HESGSL uses hierarchical embeddings to capture both local and global graph structures. It employs self-supervised learning with clustering and attention mechanisms to enhance feature representation and performance on downstream tasks.

- **SEGSL** Zou et al. (2023). SEGSL constructs a graph by fusing a kNN graph with the original graph, leveraging structural entropy and encoding tree to refine edges and hierarchically extract community structures. The graph and model parameters are optimized jointly during training.

- **ProGNN** Jin et al. (2020). ProGNN refines graph structures by enforcing low-rank, sparsity, and similarity constraints, jointly optimizing the learned graph and model parameters without predefined GSL bases or graph fusion.

- **SUBLIME** Liu et al. (2022b). SUBLIME constructs GSL graphs using an anchor view (original graph) and a learner view. The learn view is initialized with kNN and optimized with parameterized or non-parameterized methods, followed by post-processing steps like top-k filtering and symmetrization. Contrastive learning between the two views refines the graph for downstream tasks.

- **STABLE** Li et al. (2022). STABLE employs Graph Contrastive Learning (GCL) to generate robust graphs by perturbing node similarities and edges. Positive samples use slight perturbations, while negative samples use shuffled features. The graph refinement step applies a top-k filtering strategy on the node similarity matrix to retain helpful edges while removing adversarial ones.

- **CoGSL** Liu et al. (2022a). CoGSL constructs GSL graphs by combining the original graph with additional views, such as adjacency or kNN graphs. Node embeddings are generated via GCNs, and connection probabilities are estimated and refined using InfoNCE loss to maximize mutual information. The final graph is obtained through early fusion and updated with model parameters.

- **BORF** Nguyen et al. (2023). BORF is a curvature-based graph rewiring approach aimed at addressing over-smoothing and over-squashing in GNNs. It leverages Ollivier-Ricci curvature to identify edges with extreme curvature values—positive for over-smoothing and

negative for over-squashing. This process preserves the graph's topology while improving message-passing efficiency and enhancing GNN performance on downstream tasks.

- **DHGR** Bi et al. (2024). DHGR enhances GNN performance on heterophily graphs by preprocessing the graph structure through rewiring. It systematically adds homophilic edges and removes heterophilic ones based on label and feature similarities, improving node classification accuracy, particularly for heterophily settings.

- **GraphEdit** Guo et al. (2024). GraphEdit refines graph structures by integrating LLMs with a lightweight edge predictor to enhance graph representation learning. By leveraging instruction-tuning and training edge predictor, GraphEdit effectively denoises noisy connections and identifies implicit relationships among nodes.

- **LLM4RGNN** Zhang et al. (2024). LLM4RGNN is a framework that enhances the adversarial robustness of GNNs using LLMs. It detects malicious edges and restores critical ones through a purification strategy, leveraging fine-tuned LLMs for edge prediction and correction. This approach effectively mitigates topology attacks, improving GNN performance across attack scenarios.

- **LangGSL** Su et al. (2024). LangGSL presents a novel framework that integrates LLMs with GSL in an end-to-end manner. It simultaneously optimizes node features and graph connectivity by using LLMs to denoise textual attributes while GSL modules refine structural relations. This joint design enhances both semantic understanding and topological quality, leading to promising performance across diverse text-attributed graph datasets.

## I EXPERIMENTAL SETUP

For LLaTA and all baselines, we uniformly adopt 2-layer GCN as encoder, the hidden dimensional is set to 64, and dropout (rate=0.5) is applied between GCN layers. The GNN encoder is optimized for 200 epochs using the Adam optimizer, with an initial learning rate of 0.01 and weight decay of $5e$-4. We use early stopping with a patience of 20 epochs on the validation loss. For LLM-based methods, we primarily select GLM-4-9B as the LLM backbone. For LLaTA, we utilize the GLM-4 to perform inference on the node classification probabilities. For LLM4RGNN, we use the GLM-4 to construct the fine-tuning dataset and fine-tune the Llama-3-8B for edge prediction; for GraphEdit, we use the GLM-4 for both fine-tuning and edge prediction; for LangGSL, we use the GLM-4 for text cleaning and use Sentence-BERT as the LM encoder.

All experiments are repeated 10 times with different random seeds, and results are reported as $mean \pm std$. For the node clustering experiments, we adopt the $k$-means algorithm and use NAFS to evaluate the clustering performance. NAFS (Normalized Adjusted F-score) is a metric ranging from 0 to 1 that measures the consistency between predicted clusters and ground-truth labels, where higher values indicate better clustering quality. In the main text, we present the NAFS values in percentage form. We run all the experiments on NVIDIA A100 (80G memory) GPUs with CUDA 11.2 and set the time limit to 72 hours.

For LLaTA, the hyperparameters are searched as follows: The height of encoding tree $K$ in Sec. 4.1 is chosen from {2, 3, 4, 5}, the similarity threshold $\epsilon$ in Sec. 4.2 is selected from [0.4, 0.6], while the size of the candidate set $\theta$ in Sec. 4.3 is searched in {3, 5, 10, 15, 20}, and the two-step sampling frequency $r$ is tuned among {1, 3, 5, 10, 25}.

For other baselines and backbones, we primarily adopt the optimal hyperparameters reported in their original papers. If no such configurations are available, we utilize Optuna to automatically determine the best hyperparameter settings for these baselines.

## J CASE STUDY

In this section, we conduct experiments on the Citeseer, focusing on a low-level community as a case study to track and visualize the data flow within the LLaTA pipeline. Based on this, we aim to provide a clear and intuitive explanation of our method, particularly focusing on LLM inference. Our goal is to make the process transparent, thereby avoiding the uncontrollable nature of a black-box approach.

Reviewing the three modules of LLaTA (detailed descriptions can be found in Sec. 4.1-4.3), we observe that Step 1: Topology-aware In-context Construction (Sec. 4.1) and Step 3: Leaf-oriented Two-step Sampling (Sec. 4.3) have relatively fixed computation in which LLMs do not actively participate or play a secondary role. This is not directly related to our goal of visualizing the LLM inference through the case study to provide a deeper analysis of LLaTA's interpretability. Therefore, we focus on Step 2: Tree-prompted LLM Inference (Sec. 4.2), specifically analyzing its three components: *Reception-aware Leaf Augmentation*, *Community of Thought*, and *Leaf Dependency Allocation*. In Fig. 7, we use a low-level community as an example and offer a detailed demonstration of the LLaTA inference.

**Reception-aware Leaf Augmentation.** After tree initialization by minimizing SE, we aim to construct high-quality, topology-aware in-context for LLM inference. In this process, the quality of node-wise textual information is crucial, as it directly impacts the LLM's semantic understanding and the confidence of inference results. Therefore, we first perform text propagation and aggregation within each community, guided by the tree, to achieve leaf augmentation. Specifically, we input a low-level community and perform text propagation and aggregation within this community to enrich the text of each node. As illustrated in Fig. 7, we take red node 1 as an example. After text propagation and aggregation, this node acquires the text of red node 2 for enhancement. The text of red node 1 is then combined with that of red node 2 using the following prompt:

> **Prompt 1 - Part I**: Here is a paper 1 which belongs to one of the following categories: Agents, Machine Learning, Information Retrieval, Database, Human-Computer Interaction, Artificial Intelligence.
>
> The description of Agents: This research area encompasses topics such as multi-agent systems, reinforcement learning, and agent-based modeling...
>
> The description of Machine Learning: This research area focuses on developing algorithms that enable systems to learn from data and make predictions or decisions without explicit programming...
>
> The description of ...

> **Prompt 1 - Part II**: The abstract of paper 1: The Computational Theory of Neural Networks In the present paper a detailed taxonomy of neural network models with various restrictions is presented with respect to their computational properties. The criteria of classification include e.g. feedforward and recurrent architectures, discrete...

> **Prompt 1 - Part III**: The abstract of other papers related to its content: A Computational Taxonomy and Survey of Neural Network Models We survey and summarize the existing literature on the computational aspects of neural network models, by presenting a detailed taxonomy of the various models according to their computational characteristics...

**Community of Thought.** Leveraging the enhanced leaf nodes, we propose a Community of Thought (CoT) prompt mechanism, which incorporates community-enhanced descriptions for each node and its related neighbors, facilitating more reliable LLM inference. Specifically, we combine the previously obtained enhanced textual information (obtained by Prompt 1) with the community-enhanced descriptions and the specific requirements of the inference task (Prompt 2). These combined inputs are then fed into the LLM to predict the probability of the node belonging to each category. Based on the LLM's output, we apply the softmax function to obtain the node's soft label, mapping it into the semantic vector space encoded by the LLM. As an illustrative example, we present the formulation of Prompt 2, the corresponding LLM output, and the derived soft label for red node 1, as shown below.

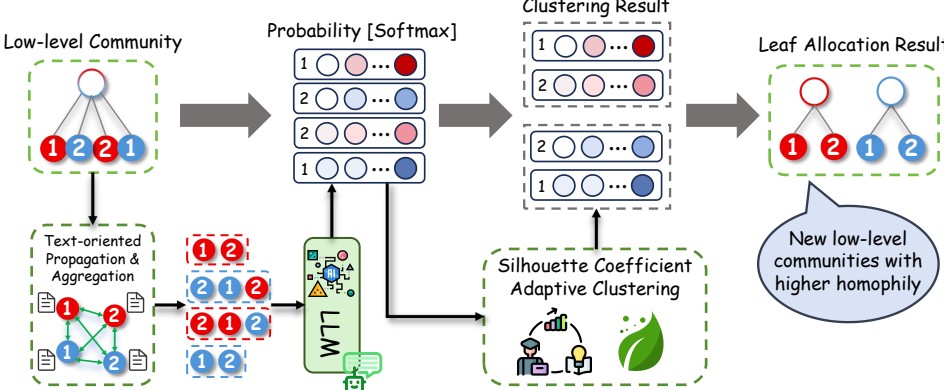

Figure 7: A case study of the tree-prompted LLaTA inference.

---

**Prompt 2**: Based on the abstract of paper 1 and other papers, please provide the probability that paper 1 belongs to each category.

For the current paper 1, please focus on the topic, methodology, keywords, and conclusions.

For the related papers, please focus on parts similar to those on paper 1.

Use integers from 0 to 9 to represent the probabilities. 0 means it is impossible to belong to that category, and 9 means it definitely belongs to that category. The example format is: [8, 4, 1, 2, 5, 3].

---

**LLM Answer:** The probabilities of paper 1 belonging to each category are: [0, 7, 0, 0, 0, 9].

---

**Soft Label of Red Node 1 [Softmax Function]:**
[0.00, 0.12, 0.00, 0.00, 0.00, 0.88].

---

**Leaf Dependency Allocation.** Finally, based on the leaf soft labels, we perform silhouette coefficient-based adaptive clustering on the four leaf nodes. As shown in Fig. 7, the clustering results indicate that the two red nodes are grouped into one cluster, while the two blue nodes are assigned to another. Following these clustering results, we reconstruct two new low-level communities to replace the original ones. This process ultimately enhances community homophily and further optimizes the tree through LLM inference.

Table 7: Performance of LLaTA with different LLM backbones.

| LLM | Pubmed | WikiCS | Instagram | History |
|---|---|---|---|---|
| Llama-2-13B | $87.79_{\pm 0.31}$ | $80.64_{\pm 0.32}$ | $66.12_{\pm 0.21}$ | $84.21_{\pm 0.28}$ |
| Llama-3-8B | $88.01_{\pm 0.33}$ | $81.13_{\pm 0.29}$ | $66.34_{\pm 0.21}$ | $84.86_{\pm 0.35}$ |
| Mistral-7B | $88.07_{\pm 0.28}$ | $81.02_{\pm 0.24}$ | $66.29_{\pm 0.20}$ | $84.98_{\pm 0.34}$ |
| GLM-4-9B | $\mathbf{88.39_{\pm 0.23}}$ | $\mathbf{81.58_{\pm 0.20}}$ | $\mathbf{66.73_{\pm 0.13}}$ | $\mathbf{85.28_{\pm 0.24}}$ |

## K  LLM BACKBONE ANALYSIS

To further answer **Q2**, we analyze the performance of LLaTA with different LLM backbones. Specifically, we evaluate the node classification performance of LLaTA with different LLM backbones on four datasets from different domains, with the results presented in Table 7.

The experimental results demonstrate that LLaTA performs effectively with different LLM backbones, highlighting the versatility of our method. GLM-4-9B outperforms the other LLM backbones,

particularly on the WikiCS and Instagram datasets, due to its stronger in-context learning ability and stable outputs, which lead to higher inference accuracy. In contrast, Llama-2-13B shows the worst performance, especially on WikiCS and History, likely due to its limited ability to grasp complex contextual relationships and output instability. Llama-3-8B and Mistral-7B provide more balanced results, with Mistral-7B offering a good trade-off between efficiency and performance. These findings emphasize that LLaTA can effectively leverage various LLM backbones for downstream tasks. Moreover, the performance differences highlight the importance of strong text inference abilities in LLMs, when combined with GSL to enhance the performance of downstream tasks. Overall, the experimental results demonstrate that LLaTA is a flexible and interpretable method, which effectively integrates LLMs with GSL.

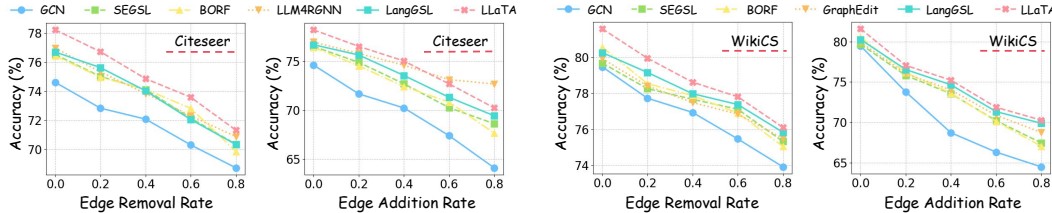

Figure 8: Node classification performance on the Citeseer and WikiCS dataset under real-world scenarios (Sparsity [Edge Removal] and Noise [Edge Addition]).

## L    ROBUSTNESS ANALYSIS

To further examine the robustness of LLaTA in practical environments, we conduct additional experiments under sparsity and noise scenarios, where the graph structure is perturbed by randomly removing or adding edges. The results on four benchmark datasets (Cora, Pubmed, Citeseer, and WikiCS) are shown in Fig. 3 and Fig. 8.

Overall, LLaTA demonstrates strong robustness across both settings. In the sparsity scenario (edge removal), LLaTA consistently outperforms all baselines across different perturbation levels. This robustness stems from our tree-based optimization, where structural entropy guides the identification of uncertain regions, ensuring that key communities remain stable even when connections are missing. As a result, the information loss caused by sparsity is effectively mitigated.

In the noise scenario (edge addition), LLaTA also achieves competitive performance. The community-aware tree prompts allow the model to suppress the influence of noisy edges by relying more heavily on semantic consistency and hierarchical community information. Nevertheless, we observe that as the edge addition rate becomes large (e.g., 0.6 or 0.8), LLaTA's performance gradually deteriorates. This phenomenon can be attributed to the irreversible negative effect of excessive noisy edges, which interfere with the initialization of the structural encoding tree. Importantly, although LLaTA remains among the top-performing methods, LLM4RGNN occasionally surpasses it under high noise rates. This is expected, since LLM4RGNN is explicitly designed to handle adversarial settings by leveraging fine-tuned LLMs for edge-level detection and correction.

In summary, these results highlight two key insights. First, LLaTA exhibits strong resilience to sparsity and moderate levels of noise, validating the effectiveness of its topology-aware and community-guided design. Second, the performance gap under severe noise suggests a promising direction for future work: integrating more powerful noise detection mechanisms into our tree-based optimization to further enhance robustness under adversarial or heavily corrupted graphs.

## M    HYPERPARAMETER ANALYSIS

To further answer **Q3**, we provide more detailed experimental results and analysis for the four hyperparameters ($K$ in Sec. 4.1, $\epsilon$ in Sec. 4.2, $\theta$ and $r$ in Sec. 4.3).

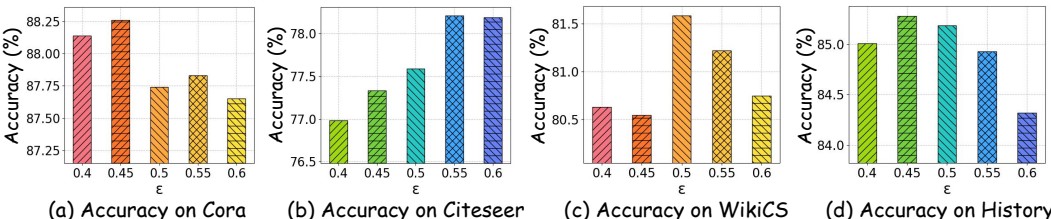

Figure 9: The node classification accuracy of LLaTA with different $\epsilon$ on Cora, Citeseer, WikiCS and History datasets.

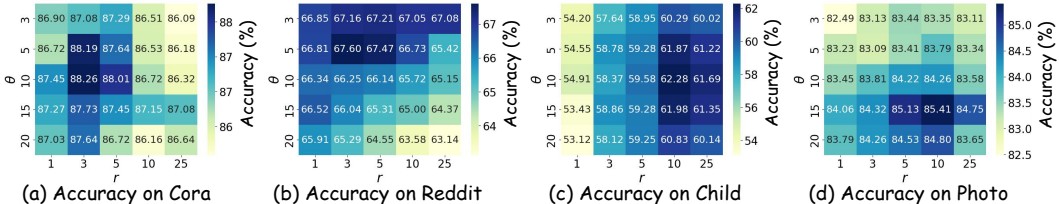

(a) Accuracy on Cora     (b) Accuracy on Reddit     (c) Accuracy on Child     (d) Accuracy on Photo

Figure 10: The node classification accuracy of LLaTA with different $\theta$ and $r$ on Cora, Reddit, Child and Photo datasets.

### M.1 ANALYSIS OF HYPERPARAMETER $K$

We further examine the effect of the encoding tree height $K$, with results shown in Fig. 4 (left). The experiments reveal that the optimal $K$ varies across datasets, reflecting differences in graph structural complexity. Moderate depths ($K = 2$ or $3$) generally achieve the best performance, as they strike a balance between capturing multi-level community information and avoiding redundant hierarchy. However, as $K$ increases beyond 5, accuracy consistently drops for all datasets, and the decline becomes pronounced at $K = 6$. This suggests that excessively deep trees may dilute meaningful community structures and introduce noise, thereby harming the quality of structural optimization. Overall, these findings highlight that $K$ should be tuned according to graph complexity, while overly large values should be avoided due to their adverse effect on performance.

### M.2 ANALYSIS OF HYPERPARAMETER $\epsilon$

We further study the effect of $\epsilon$, which controls the threshold for constructing in-context information, on four datasets: Cora, Citeseer, WikiCS, and History, as shown in Fig. 9.

The results confirm that $\epsilon$ plays a crucial role in balancing noise reduction and information preservation. When $\epsilon$ is too small (e.g., $\epsilon = 0.4$), nodes from different classes are aggregated, introducing noisy contexts and leading to reduced inference accuracy. In contrast, when $\epsilon$ becomes too large (e.g., $\epsilon = 0.6$), the model over-filters, discarding meaningful connections and thereby losing valuable semantic and structural cues. Across datasets, the optimal $\epsilon$ varies slightly depending on graph characteristics: Cora and History achieve peak performance at $\epsilon = 0.45$, Citeseer at $\epsilon = 0.55$, and WikiCS at $\epsilon = 0.5$. Despite these variations, the overall performance remains stable within the range $[0.45, 0.55]$, suggesting that this interval provides a reliable trade-off between minimizing cross-class noise and preserving intra-class relationships.

In summary, $\epsilon$ is a sensitive yet well-bounded hyperparameter: choosing $\epsilon \in [0.45, 0.55]$ generally yields robust performance across both homophilic and heterophilic graphs. This validates the effectiveness of our design in constructing high-quality in-context information for LLMs, while also offering practitioners practical guidance for parameter tuning in diverse real-world scenarios.

### M.3 ANALYSIS OF HYPERPARAMETERS $\theta$ AND $r$

To provide a comprehensive understanding of how $\theta$ (semantic sampling size) and $r$ (two-step sampling frequency) influence LLaTA, we conduct experiments on five representative datasets, with results summarized in Fig. 4 and Fig. 10.

**Impact of $r$.** We observe that $r$ exerts a stronger dataset-dependent effect. On heterophilic graphs such as Child, accuracy is highly sensitive to the choice of $r$: performance steadily improves up to $r = 10$, but drops when $r$ is further increased (e.g., $r = 25$). This degradation arises because excessive edge additions overwrite useful heterophilic patterns, forcing the structure toward artificial homophily. By contrast, on homophilic datasets such as History, Cora, and Photo, performance remains stable across a wide range of $r$ values, with fluctuations typically within 0.5%. These results indicate that while heterophilic graphs require careful calibration of $r$ to preserve structural integrity, homophilic graphs tolerate larger $r$ without substantial accuracy loss.

**Impact of $\theta$.** The semantic sampling parameter $\theta$ exhibits remarkable stability across all datasets. Within the range $[5, 15]$, performance variations remain minor, typically under 1%. Extreme settings, however, show performance degradation: when $\theta = 3$, the sampling space becomes too restrictive, limiting semantic coverage; when $\theta = 20$, excessive candidate expansion introduces noise, weakening the reliability of semantic similarity. Importantly, these effects are consistent across homophilic and heterophilic datasets, demonstrating that $\theta$ primarily influences the quality of semantic sampling rather than the structural optimization itself.

From these findings, two guidelines emerge: (1) $r$ should be prioritized during tuning, especially for heterophilic graphs where improper calibration may lead to over-homogenization of node neighborhoods; (2) $\theta \in [5, 15]$ can be regarded as a reliable default choice for diverse graph types, providing a balance between semantic coverage and efficiency. Taken together, these results not only validate the robustness of LLaTA across varying parameter settings but also highlight its adaptability: while $r$ controls the sensitivity to graph topology, $\theta$ primarily refines semantic discrimination. This complementary role of the two hyperparameters further underscores the stability of our framework under both homophilic and heterophilic conditions.

## M.4    GENERAL GUIDANCE ON HYPERPARAMETER SELECTION

Here we provide additional guidelines for selecting the four key hyperparameters in LLaTA, based on both theoretical considerations and empirical observations:

**Encoding Tree Height $K$.** As discussed in Sec. 4.1, the height of the encoding tree determines the number of hierarchical community divisions. Its optimal value is closely related to the graph scale and the number of node categories. Larger graphs with more categories typically exhibit more complex hierarchical structures, thus requiring deeper trees. Empirically, we observe that the optimal $K$ usually falls within the range of 2 to 5 for most datasets.

**Threshold $\epsilon$.** The threshold $\epsilon$ governs text propagation and aggregation. Its optimal setting depends on the quality of both textual information and the encoding tree. When either is of low quality, a higher $\epsilon$ enforces stricter filtering, thereby reducing noise and preventing unreliable information from propagating. With higher-quality inputs, smaller $\epsilon$ values can be adopted to retain richer contextual information.

**Candidate Set Size $\theta$.** The parameter $\theta$ controls the size of the candidate set for semantic similarity sampling. Its optimal value depends on both the quality of node text and the original graph structure. When textual attributes and structural quality are high, LLM-generated semantic embeddings are more reliable, allowing for a larger $\theta$ to improve coverage. Conversely, lower-quality data may benefit from smaller $\theta$ values to reduce noise.

**Sampling Frequency $r$.** The frequency $r$ in two-step sampling is influenced by the average degree of the original and optimized graphs. For graphs with higher average degrees, larger $r$ values are generally needed to preserve sufficient structural connectivity. In contrast, for sparse graphs, moderate $r$ values are preferable to avoid excessive edge additions.

These guidelines offer practical insights for tuning LLaTA across diverse graph types, providing a balance between general applicability and dataset-specific adaptability.

Table 8: Complete runtime(h) results of LLaTA across all 11 datasets.

| Method | Cora | Citeseer | Pubmed | WikiCS | Instagram | Reddit | Ratings | Child | History | Photo | ArXiv |
|---|---|---|---|---|---|---|---|---|---|---|---|
| LLaTA$_{GCN}$ | 1.593 | 1.974 | 12.873 | 7.081 | 7.850 | 21.136 | 15.583 | 15.016 | 26.394 | 30.372 | 53.175 |
| LLaTA$_{GAT}$ | 1.598 | 1.980 | 12.881 | 7.124 | 7.861 | 21.145 | 15.589 | 15.025 | 26.451 | 30.468 | 53.479 |
| LLaTA$_{SAGE}$ | 1.583 | 1.958 | 12.861 | 7.076 | 7.847 | 21.131 | 15.578 | 15.009 | 26.489 | 30.369 | 53.177 |

## N  EFFICIENCY ANALYSIS OF LLaTA

In this section, we present the complete runtime performance of LLaTA across 11 tagged datasets in Table 8, along with a comprehensive analysis of our method's computational efficiency advantages.

Current graph structure learning (GSL) methods face significant efficiency bottlenecks, particularly in large-scale or noisy graphs. Traditional approaches—including metric learning, probabilistic modeling, and direct optimization—often require computationally expensive pairwise similarity calculations (e.g., kernel functions or attention mechanisms) or iterative optimization of adjacency matricesZhu et al. (2021). These operations exhibit quadratic or higher complexity relative to node counts, limiting scalability. For LLM-based GSL Methods (e.g., GraphEdit Guo et al. (2024), LLM4RGNN Zhang et al. (2024)), efficiency issues are further exacerbated by the inherent overhead of large-model inference and fine-tuning. Empirical results reveal that these methods exceed 72 hours on larger datasets among the 11 benchmarks (As shown in Table 1), primarily due to their reliance on full-parameter tuning and dense graph operations.

Compared to existing paradigms, our framework achieves superior efficiency by: **(a) Topology-Aware Context Enhancement**: The Community-of-Thought mechanism provides high-quality topological context to guide LLM reasoning, eliminating the need for fine-tuning. By leveraging the inherent structure-awareness of communities, we reduce the dependency on costly end-to-end training while improving inference accuracy. **(b) Parameter-Free Structure Reconstruction**: A tree-based sampler enables lightweight graph restructuring without trainable parameters. This avoids the computational overhead of gradient-based adjacency matrix optimization (e.g., nuclear norm regularization in ProGNN Jin et al. (2020)) or probabilistic sampling (e.g., Gumbel-Softmax in NeuralSparse Zheng et al. (2020)).

**Furthermore**, we provide an **efficiency-oriented acceleration scheme** for large-scale graphs to further enhance the scalability of our method. To reduce computational overhead while maintaining the effectiveness of structure optimization, we introduce a **composite scoring mechanism** to identify the most needed low-level communities for optimization.

$$\text{Score}(\mathcal{C}^{\ell}) = \lambda_1 \cdot \mathcal{H}_{\text{struct}}(\mathcal{C}^{\ell}) + \lambda_2 \cdot \mathcal{H}_{\text{label}}(\mathcal{C}^{\ell}) + \lambda_3 \cdot \left(1 - \mu_{\text{textsim}}(\mathcal{C}^{\ell})\right), \tag{41}$$

where $\mathcal{H}_{\text{struct}}(\mathcal{C}^{\ell})$ denotes the structural entropy, quantifying topological complexity; $\mathcal{H}_{\text{label}}(\mathcal{C}^{\ell})$ measures label distribution entropy, capturing heterogeneity; and $\mu_{\text{textsim}}(\mathcal{C}^{\ell})$ is the mean pairwise semantic similarity within the community. Thus, $1 - \mu_{\text{textsim}}(\mathcal{C}^{\ell})$ reflects semantic inconsistency, assigning higher scores to communities with lower internal similarity. By ranking communities with this score, we prioritize optimization and sampling on top-ranked candidates. This targeted strategy significantly reduces computational overhead while preserving the effectiveness of structure optimization, enabling LLaTA to scale to large datasets such as ArXiv within practical time limits. In our experiments on the arXiv dataset, we applied the community sampling algorithm to select 40% of the communities for **Tree-prompted LLM Inference** process, which significantly improved efficiency on large-scale graphs while preserving the effectiveness of our method.

**(c) Efficiency-Oriented Community Selection**: The composite scoring mechanism adaptively identifies and optimizes only the most informative communities, reducing redundant computation and ensuring scalability to large graphs.

Enabled by these design choices, our framework achieves superior efficiency through the following three aspects: **(1) Eliminating Fine-Tuning**: By completely bypassing parameter updates of LLMs, our method reduces both GPU memory footprint and training time by orders of magnitude. This makes the framework applicable to resource-constrained environments where fine-tuning large

models is infeasible. **(2) Sublinear Complexity**: The tree-based sampler operates locally within communities rather than across the entire graph, effectively reducing global pairwise computations. This results in sublinear complexity with respect to the number of nodes, making our method scalable to large and dense graphs. **(3) Parallelizability and Efficiency-Oriented Community Selection**: Community-level operations are inherently parallelizable and can be distributed across multiple processors, enabling scalability to graphs with $10^{5+}$ nodes where prior LLM-GSL methods fail. Moreover, our composite scoring mechanism further enhances scalability by adaptively identifying and optimizing only the most promising communities, thereby reducing redundant computation while maintaining the effectiveness of structure optimization (e.g., on the arXiv dataset we optimize only 40% of the communities while achieving competitive performance).

## O  USE OF LARGE LANGUAGE MODELS IN PAPER WRITING

In preparing this manuscript, we made use of generative artificial intelligence (GenAI) tools, specifically GPT-4o and GPT-5, to assist with text refinement and polishing, as well as with the drafting and modification of auxiliary code snippets. These tools were employed solely to enhance clarity and readability and to streamline the presentation of supporting materials. Importantly, GenAI was not involved in the derivation of mathematical formulas, the design or implementation of core algorithms, or the development of key scientific insights. All theoretical analyses, algorithmic contributions, and experimental validations were carried out independently by the authors to ensure the originality and integrity of the work. Furthermore, all AI-assisted outputs were rigorously reviewed and verified by the authors to guarantee their accuracy and consistency with the scientific content, thereby upholding the reliability of the results presented.

