# OpenReview forum: "Unlocking Graph Structure Learning with Tree-Guided Large Language Models"
_ICLR.cc/2026/Conference — ICLR 2026 Conference Withdrawn Submission_

### Official Review · Reviewer_pPmG · 2025-10-31

**Soundness:** 3
**Presentation:** 2
**Contribution:** 3
**Rating:** 6
**Confidence:** 4

**Summary:**

The paper proposes LLaTA, a tree-guided framework for graph structure learning (GSL) that leverages large language models (LLMs) through in-context inference rather than fine-tuning. It first constructs a hierarchical structural-entropy tree to represent multi-level communities, then uses LLM-based semantic similarity to refine community assignments and modify graph edges. This process produces an improved, text-aware graph structure that is later used for downstream GNN tasks such as node classification and clustering. Experiments on multiple text-attributed graph datasets show accuracy gains over prior GSL methods, though the improvement may largely reflect the LLM’s predictive bias rather than genuine structural learning.

**Strengths:**

1. The paper introduces a tree-based optimization framework that replaces traditional edge-predictor training with a language-aware structural entropy tree, offering a fresh and interpretable perspective on how to integrate topology and textual semantics.

2. Unlike prior LLM–GNN hybrids that rely on instruction fine-tuning, this approach leverages in-context learning for structure refinement, demonstrating a more computationally efficient and flexible use of LLMs across diverse graph backbones.

3. The authors conduct experiments on 11 text-attributed graph datasets with 14 baselines, showing consistent improvements in both node classification and clustering tasks, and providing a broad empirical basis for comparison.

**Weaknesses:**

1. The paper’s presentation is overly complicated, relying on dense mathematical notation and cluttered figures that obscure the main idea. The core pipeline could be described much more clearly and intuitively. For instance, Equation (3) merely represents the cosine similarity between two feature vectors, which could be explained directly in words rather than presented as a full formula. Similarly, Figures 1 and 2 contain overlapping and overly detailed content; the authors are encouraged to reorganize them, perhaps simplifying Figure 1 to convey the overall framework while using Figure 2 to emphasize the key procedural details.

2. The performance improvement may largely stem from the LLM’s predictive bias rather than the proposed structural-learning mechanism. Since the model reassigns nodes to communities based on LLM-predicted labels, the gain could simply result from increased label homophily, not genuine structural enhancement.

**Questions:**

1. I am curious about the accuracy of the soft labels predicted by the LLM. Could the authors report these metrics and compare them with LLaTA’s final performance? This would help clarify how much of the overall improvement originates from the LLM’s predictive capability versus the subsequent structural refinement.

---

### Official Review · Reviewer_MuF8 · 2025-11-01

**Soundness:** 2
**Presentation:** 2
**Contribution:** 1
**Rating:** 2
**Confidence:** 5

**Summary:**

This paper introduces LLaTA, a framework that integrates large language models into graph structure learning (GSL) for text-attributed graphs. It reformulates GSL as a tree optimization problem, focusing on language-aware tree sampling rather than traditional edge prediction, and adopts a decoupled, training-free design for efficient LLM integration. Experiments show that LLaTA achieves superior performance.

**Strengths:**

1. Graph structure learning on TAGs is underexplored and deserves more attention from the community.
2. The overall framework seems sound and reasonable.

**Weaknesses:**

1. The writing is overwhelming, with substantial concepts and equations scattered throughout. Going through the paper, it is difficult to grasp the core contribution of the work, and it thus seems like a combination of existing methods and engineering trial-and-error.

2. Efficiency is a major concern. Although the paper claims to be the most efficient among LLM-enhanced GSL methods, it still incurs substantial costs, e.g., tree construction and LLM inference. Table 8 shows that even a small graph like Cora requires over 1.5 hours of runtime, while ArXiv involves inference exceeding 50 hours. In contrast, traditional GNN methods and LLM-free GSL methods converge within a few minutes. It is highly arguable whether the reported gains are worth the cost, especially given the marginal improvements on many datasets.

3. The paper lacks key baselines: it does not compare with traditional GNN methods that do not include any structure learning modules. Prior work suggests that GSLs often fail to improve over simple GNNs [1]. It would be important to show the actual gains of the proposed method compared with GSL-free baselines.

4. Following the above, the reported gains might primarily come from the use of LLMs to process textual features rather than from structure refinement itself. It would therefore be important to include GNN–LLM integration baselines such as GIANT[2] and GLEM [3].

[1] https://arxiv.org/pdf/2310.05174

[2] https://arxiv.org/pdf/2111.00064

[3] https://arxiv.org/pdf/2210.14709

**Questions:**

What steps can be taken to significantly reduce the cost of LLM-enhanced GSL methods so that they become feasible for real-world deployment?

---

### Official Review · Reviewer_adfX · 2025-11-01

**Soundness:** 3
**Presentation:** 1
**Contribution:** 2
**Rating:** 4
**Confidence:** 4

**Summary:**

This paper proposes LLaTA, a training-free framework for Graph Structure Learning on text-attributed graphs. The key idea is to reformulate GSL as a tree-guided optimization problem using structural entropy to build hierarchical tree representations of graphs, and then use LLMs to reason over these trees through in-context prompting.

**Strengths:**

1. The paper addresses an interesting problem: leveraging LLMs for structure learning on text-attributed graphs, a direction that is increasingly important.

2. Testing on 11 diverse datasets across multiple domains with various baselines demonstrates thoroughness.

3. The use of structural entropy for tree construction provides interpretability and is well-grounded in information theory.

**Weaknesses:**

1. The technical contribution is rather incremental. Much of LLaTA is built upon existing structural entropy–based tree construction (e.g., SE-GSL), and the LLM component mainly serves as an in-context inference layer. The method appears more like a combination of known techniques than a fundamentally new algorithmic paradigm. The authors should highlight what truly differentiates LLaTA beyond simply merging SE-GSL and LLM prompting.

2. While each component is understandable in isolation, the writing seems intentionally obfuscated. The overall connection between them is not clearly explained. The paper reads like a list of loosely related techniques rather than a unified model. This weakens the narrative and makes the motivation blurry.

3. The paper is verbose and somewhat redundant, making it difficult to follow. Figures (e.g., Fig. 2) contain too much text, reducing readability. Moreover, most experimental details are deferred to the appendix, making it hard to assess the setup from the main text. A cleaner and more focused presentation is necessary.

4. Although the method is training-free, the LLM inference step dominates runtime, especially on large graphs. Moreover, the reported results are not particularly strong — for instance, the performance on OGBN-Arxiv is lower than some older models [1,2,3] that used LMs in stead of LLMs. This questions the claimed efficiency and scalability.

5. Some important baselines are omitted in Table 1. For example, although GLEM appears in the Table 4 and could achieve better performance on several datasets such as arxiv but is not compared in Table 1.

6. According to the analysis of complexity, the computational cost of structural entropy–based tree construction is nontrivial and could dominate overall runtime on large graphs. The paper does not report any performance or efficiency analysis on truly large-scale datasets such as OGBN-Papers100M or MAG240M.

7. The paper only reports results on node classification and clustering, despite claiming general applicability to “any backbone and any downstream task.” There are no link prediction or graph-level tasks, which are standard in GSL benchmarks. This makes it difficult to evaluate the generality of the proposed approach.

8. I don't quite agree with the comments in Section 3.2: the decoupled methods are better than the coupled ones. Since the gradients of downstream tasks will not influence LLMs during fine-tuning in the decoupled model, it will make LLMs lack the information about the downstream task from back propagation.

[1] Node Feature Extraction by Self-Supervised Multi-scale Neighborhood Prediction

[2] Learning on Large-scale Text-attributed Graphs via Variational Inference

[3] Efficient End-to-end Language Model Fine-tuning on Graphs

**Questions:**

Please refer to the weakness

---

### Official Review · Reviewer_Pyy6 · 2025-11-09

**Soundness:** 3
**Presentation:** 2
**Contribution:** 2
**Rating:** 4
**Confidence:** 3

**Summary:**

This paper introduces LLaTA (Large Language and Tree Assistant), a training-free framework for graph structure learning (GSL) on text-attributed graphs (TAGs) that integrates large language models (LLMs) via a tree-based optimization paradigm.
Rather than training an edge predictor, the authors leverage structural entropy to construct a hierarchical encoding tree that captures multi-level topological patterns. This tree serves as a prompt for LLMs to perform in-context inference combining both topology and text.
Graph structures are refined through leaf dependency reallocation and a two-step sampling process guided by structural entropy and semantic similarity between LLM-generated soft labels. Experiments on 11 datasets demonstrate that LLaTA achieves state-of-the-art accuracy, strong efficiency by removing fine-tuning, scalability to large graphs where prior methods fail, and robustness under varying graph sparsity and noise conditions.

**Strengths:**

* Reformulates GSL from an edge predictor learning problem into a tree-based optimization paradigm. Introduces a principled use of structural entropy to connect topology and LLM prompting.

* Training-free and Efficiency: Avoids fine-tuning LLMs; relies purely on in-context learning. Achieves strong results with zero training cost on the graph learner.

* Strong Empirical Results: Consistent improvements across 11 datasets and 14 baselines. Demonstrates scalability where others fail (e.g., History, Photo, ArXiv datasets). Extensive ablation, robustness, and hyperparameter analysis.

**Weaknesses:**

* **Incremental Novelty**
   While the proposed tree-guided integration of structural entropy and LLM prompting is an interesting direction, the overall contribution is more like an incremental combination of existing ideas than a fundamentally new paradigm for graph structure learning. The framework largely builds on well-known entropy-based clustering and in-context reasoning principles, which limit the technical novelty of the work.

* **Limited Theoretical Insight and Missing Link to Empirical Superiority**
   Although the paper provides several theoretical results (e.g., on structural entropy and semantic similarity), these analyses do not directly justify the necessity or superiority of adopting a tree-based structure. The theory mainly formalizes general intuitions but does not explain why the proposed tree-guided formulation should outperform other methods, such as random walk–based or 1-hop neighbor aggregation. As shown in Table 3, the empirical results indeed indicate that using the tree structure yields better performance than the random walk or 1-hop variants, but this advantage currently lacks theoretical grounding or interpretive analysis in the main text. Establishing a clearer theoretical rationale for the observed empirical gain would substantially strengthen the contribution of this paper.

* **Limited Evidence on Scalability and Efficiency**
   Although the paper claims scalability and Efficiency, the runtime evaluations are restricted to very small datasets (Cora, Citeseer).

**Questions:**

* Which LLM (model, version, size) is used for inference? How sensitive is performance to the LLM’s scale or domain?

* Are the prompts (CoT) manually engineered or automatically generated?

* What is the definition of OOM and OOT in Table 1? What causes OOM/OOT in other baselines (especially non-LLM ones like BORF or SEGSL)?

* In the claimed complexity of the proposed method, what is the practical magnitude of \(t_{LLM-i}\)?

* In Table 3, what exactly are “TO”, “TO_LLM”? Could the author please further elaborate on their difference?

* What is the principle of selecting datasets for different parts of experiments?

---

### Note · Authors · 2025-11-13

I have read and agree with the venue's withdrawal policy on behalf of myself and my co-authors.